# Sirtuin5 protects colorectal cancer from DNA damage by keeping nucleotide availability

Hao-Lian Wang[1,3], Yan Chen[1,3], Yun-Qian Wang[1,3], En-Wei Tao[1], Juan Tan[1], Qian-Qian Liu[1], Chun-Min Li[1], Xue-Mei Tong [2], Qin-Yan Gao[1], Jie Hong[1], Ying-Xuan Chen [1] ✉ & Jing-Yuan Fang[1]

In our previous study, we reported that sirtuin5 (SIRT5), a member of the NAD⁺-dependent class III histone deacetylase family, is highly expressed in colorectal cancer (CRC). Herein we show that SIRT5 knockdown impairs the production of ribose-5-phosphate, which is essential for nucleotide synthesis, resulting in continuous and irreparable DNA damage and consequently leading to cell cycle arrest and enhanced apoptosis in CRC cells. These SIRT5 silencing-induced effects can be reversed by nucleoside supplementation. Mechanistically, SIRT5 activates transketolase (TKT), a key enzyme in the non-oxidative pentose phosphate pathway, in a demalonylation-dependent manner. Furthermore, TKT is essential for SIRT5-induced malignant phenotypes of CRC both in vivo and in vitro. Altogether, SIRT5 silencing induces DNA damage in CRC via post-translational modifications and inhibits tumor growth, suggesting that SIRT5 can serve as a promising target for CRC treatment.

Metabolic reprogramming is one of the most important hallmarks of various types of cancers, including colorectal cancer (CRC), which has the second highest mortality rate worldwide[1]. Sirtuin5 (SIRT5) belongs to the family of nicotinamide adenine dinucleotide (NAD⁺)-dependent class III histone deacetylase enzymes; it is found in mitochondria as well as cytosol, and reportedly regulates diverse metabolic pathways[2,3]. Although SIRT5 was initially considered to be a deacetylase[4], it was recently found to have potent lysine demalonylase[2,5], desuccinylase[6,7], and deglutarylase[8,9] activities. Accumulating evidence suggests that these non-canonical post-translational modifications (PTMs) are involved in the regulation of cancer metabolic adaptations[10]. A previous study showed that the demalonylation of succinate dehydrogenase complex subunit A by SIRT5 led to succinate accumulation, resulting in the activation of thioredoxin reductase 2 and resistance to chemotherapy[11]. Moreover, SIRT5 evidently protects glutaminase from ubiquitin-mediated degradation in a desuccinylation-dependent manner and elevates carbon and/or nitrogen levels, thereby promoting breast tumor tumorigenesis[12]. Considering that SIRT5 promotes cancer cell survival and proliferation in a context-specific manner[12–14],

its role in the metabolic reprogramming of tumors needs to be comprehensively explored.

A recent study suggested that altered metabolism affects genome stability related to DNA damage[15]. Genomic DNA, the main carrier of genetic material, is under constant attack from exogenous as well as endogenous DNA damaging agents, which can result in replication errors[16]. In the absence of prompt and accurate DNA repair mechanisms, cell senescence, cell cycle arrest, and even cell apoptosis may occur[17]. Intriguingly, tumor cells can protect themselves from unfavorable changes, including DNA damage, in the tumor microenvironment via metabolic reprogramming. They have been reported to use glutamine for anaplerosis, i.e., refueling the pool of precursor molecules to maintain cell growth[18]. In rapidly proliferative tumor cells, the pentose phosphate pathway (PPP) is an important source of ribose-5-phosphate (R5P) and nicotinamide adenine dinucleotide phosphate (NADPH), which are chief precursors and hydrogen donors for DNA and RNA biosynthesis, respectively[19]. In general, the PPP is composed of two main branches: (1) oxidative, generating R5P, NADPH, and $CO_2$ from glucose-6-phosphate and (2) non-oxidative, converting the

[1]State Key Laboratory for Oncogenes and Related Genes, Division of Gastroenterology and Hepatology, Renji Hospital, School of Medicine, Shanghai Jiao Tong University, Shanghai, China. [2]Department of Biochemistry and Molecular Cell Biology, Shanghai Key Laboratory for Tumor Microenvironment and Inflammation, Key Laboratory of Cell Differentiation and Apoptosis of Chinese Ministry of Education, Shanghai Jiao Tong University School of Medicine, Shanghai, China. [3]These authors contributed equally: Hao-Lian Wang, Yan Chen, Yun-Qian Wang. ✉e-mail: yingxuanchen71@sjtu.edu.cn

intermediate products of glycolysis, such as fructose-6-phosphate and glyceraldehyde-3-phosphate, to R5P. Although R5P could be generated via the oxidative PPP, > 80% of R5P required by tumor cells for nucleotide synthesis is supplied by the non-oxidative PPP[20]. To date, the transcriptional regulation of the PPP mainly focuses on oncogenic mutations or alterations, such as RAS, mTOR, and NRF2[21-23]. In our previous study, we reported that SIRT5 is overexpressed in CRC tissues. Furthermore, SIRT5 knockdown in CRC cells led to cell cycle arrest and apoptosis, and SIRT5 was found to contribute to colorectal carcinogenesis by enhancing glutaminolysis in a glutarylation-dependent manner[24]. Nevertheless, little remains known about the other mechanisms underlying SIRT5 in tumors.

Herein we found that SIRT5 silencing induced DNA damage, cell cycle arrest, and cell apoptosis in CRC. Further, SIRT5 knockdown reduced R5P production by inhibiting the non-oxidative PPP, as indicated by mass spectrometry (MS) and $^{13}$C-based metabolic flux analyses. Supplementation with nucleosides rescued DNA damage, reversing cell cycle arrest and apoptosis as well as the inhibition of colony formation induced by SIRT5 silencing. Mechanistically, SIRT5 activated transketolase (TKT), a key enzyme in the non-oxidative PPP, in a demalonylation-dependent manner. We believe that these results highlight a mechanism underlying the action of SIRT5 and can improve the treatment of CRC.

## Results

### SIRT5 silencing induces DNA damage in human CRC cells

In our earlier study, we reported that SIRT5 silencing induced cell cycle arrest and apoptosis in CRC cells[24]. Interestingly, stress-induced DNA damage has also been reported to cause cell cycle arrest and apoptosis[25]. Therefore, herein we examined whether SIRT5 silencing induces DNA damage. As anticipated, the expression of γH2AX, a marker of DNA damage, was upregulated in multiple human CRC cell lines (HCT116, LoVo, and HT29) after treatment with two short-interfering RNAs (siRNAs) targeted at SIRT5 for 24, 36, 48, and 72 h (Fig. 1a and Supplementary Fig. 1a). Further, after pretreatment with caspase inhibitors z-VAD-FMK (z-VAD), γH2AX nuclear foci was increased in SIRT5-knockdown cells, indicating that SIRT5 knockdown itself, but not through apoptosis, could also cause significant DNA damage (Fig. 1b c, and Supplementary Fig. 1b, c). However, SIRT5-seemed to have little effect on the levels of γH2AX in the human normal colon epithelial cell line NCM460 (Supplementary Fig. 1a). We then used the alkaline comet assay to directly observe the effects of SIRT5 silencing on DNA damage. Consistent with other findings, a significant increase in the tail moment was observed after SIRT5 siRNAs transfection for 48 h (Fig. 1d, e). To investigate the mechanism underlying DNA damage, we measured the levels of RPA phosphorylation (pRPA), a common marker of replication stress that is accumulated in nuclear speckles at sites of stalled replication[26], using immunofluorescence and Western blotting, with hydroxyurea as a positive control. We observed that pRPA levels were increased in nuclear speckles after SIRT5 knockdown (Fig. 1f-h). Furthermore, we used DNA fiber assay to examine the impact of SIRT5 knockdown on DNA replication forks and found that SIRT5 deficiency slowed down the progression of replication forks (Supplementary Fig. 1d, e), indicating that SIRT5 silencing-induced DNA damage was associated with replication stress. Eukaryotic cells usually activate ataxia-telangiectasia mutated (ATM)/CHK2 and ATM- and RAD3-related (ATR)/CHK1 signalling pathways in response to stress-induced DNA damage to arrest the cell cycle and initiate DNA repair[27]. Herein we found an increase in the expression levels of p-ATM, p-ATR, p-CHK1, and p-CHK2 in HCT116 and LoVo cells treated with SIRT5 siRNAs for 48 h, indicating that DNA damage response (DDR) was simultaneously activated (Fig. 1i). Collectively, these results indicated that SIRT5 silencing induced DNA damage and activated DDR in human CRC cells.

### SIRT5 sustains the nucleotide pool by enhancing R5P synthesis to maintain DNA stability

Insufficient nucleotide pool can lead to replication fork stalling in cancer cells and induce double-strand breaks[28]. Besides, Patra et al. reported that cancer cells utilize the reprogramming of the PPP to replenish the pool of R5P, the precursor for the biosynthesis of all types of nucleotides[19]. To assess whether SIRT5 has an effect on metabolic reprogramming in CRC, we re-analyzed our previous GC–MS data obtained from HCT116 cells treated with SIRT5 siRNAs[29]. Metabolite set enrichment analyses revealed that SIRT5 silencing caused significant changes in the PPP (Supplementary Fig. 2a), with a marked decrease in the levels of R5P and ribulose-5-phosphate (Ru5P) (Fig. 2a and Supplementary Fig. 2b). Consistently, SIRT5 knockdown led to decreased levels of purine nucleotides, such as inosine monophosphate, adenosine monophosphate, and guanosine monophosphate. However, there were negligible changes in the levels of adenine and guanine (Supplementary Fig. 2b), suggesting that purine nucleotide deficiency was principally caused by the decrease in R5P. These results also revealed a deficiency of pyrimidine nucleotides, including uridine monophosphate (UMP) and cytidine monophosphate (CMP), in SIRT5-deficient HCT116 cells (Supplementary Fig. 2b). Interestingly, carbamoyl aspartic acid levels showed a twofold increase (Supplementary Fig. 2b), indicating that the conversion of carbamoyl aspartic acid to UMP was inhibited due to R5P deficiency, which in turn led to the abnormal synthesis of UMP and CMP in SIRT5-deficient HCT116 cells. Furthermore, targeted metabolomic analysis for nucleotides confirmed that the nucleotide pool was decreased when SIRT5 was downregulated (Fig. 2b). Considering that cancer cells tend to adapt metabolic flux to restore nucleotide pool and avoid DNA damage, we subsequently examined metabolic changes in CRC cells with stable knockdown of SIRT5. HCT116 stable cell lines were established using lentiviral particles carrying non-target control (NTC) short hairpin RNA (shRNA) or SIRT5 shRNAs. As evident from Supplementary Fig. 2c, SIRT5 shRNAs efficiently knocked down SIRT5 expression. Consistent with previous results on SIRT5 siRNAs, non-targeted ultrahigh performance liquid chromatography (UHPLC)–HRMS/MS metabolomics revealed a decrease in the levels of R5P, Ru5P, and nucleotides in HCT116 cells stably transfected with SIRT5 shRNAs (Supplementary Fig. 2d). Furthermore, targeted metabolomics confirmed R5P downregulation and nucleotide pool deficiency when SIRT5 was stably downregulated (Supplementary Fig. 2e). Collectively, these data suggested that SIRT5 silencing suppressed the PPP, impairing the production of R5P, which is essential for nucleotide synthesis.

In the absence of sufficient nucleotide levels, cancer cells cannot maintain normal DNA replication and consequently experience DNA damage. Bester et al. reported that exogenously supplied nucleosides reduced DNA damage[30]. We thus investigated the effects of exogenous supplementation of nucleosides (A, U, C, and G) on DNA damage in SIRT5-knockdown CRC cells, and found that SIRT5 silencing-induced DNA damage was reversed in a dose-dependent manner (Fig. 2c). Moreover, nucleoside supplementation led to a decrease in the number of γH2AX nuclear foci (Fig. 2d, e) and tail moment (Fig. 2f, g) in both HCT116 and LoVo cells with SIRT5 silencing. Insufficient nucleotide levels can elicit replication stress[31,32]. In line with this, we found that increased pRPA levels and decreased replication fork speed induced by SIRT5 knockdown were recovered by nucleoside supplementation (Fig. 2h and Supplementary Fig. 2f), suggesting that nucleotide deficiency is accountable for replication stress-induced DNA damage in SIRT5-knockdown CRC cells. Oxidative stress is one of the most common endogenous sources of DNA damage. Although we found that SIRT5 deficiency was accompanied by an increase in the levels of reactive oxygen species (ROS), there was no change in the levels of 8-hydroxy-20-deoxyguanosine (8-OHdG), which is a marker of

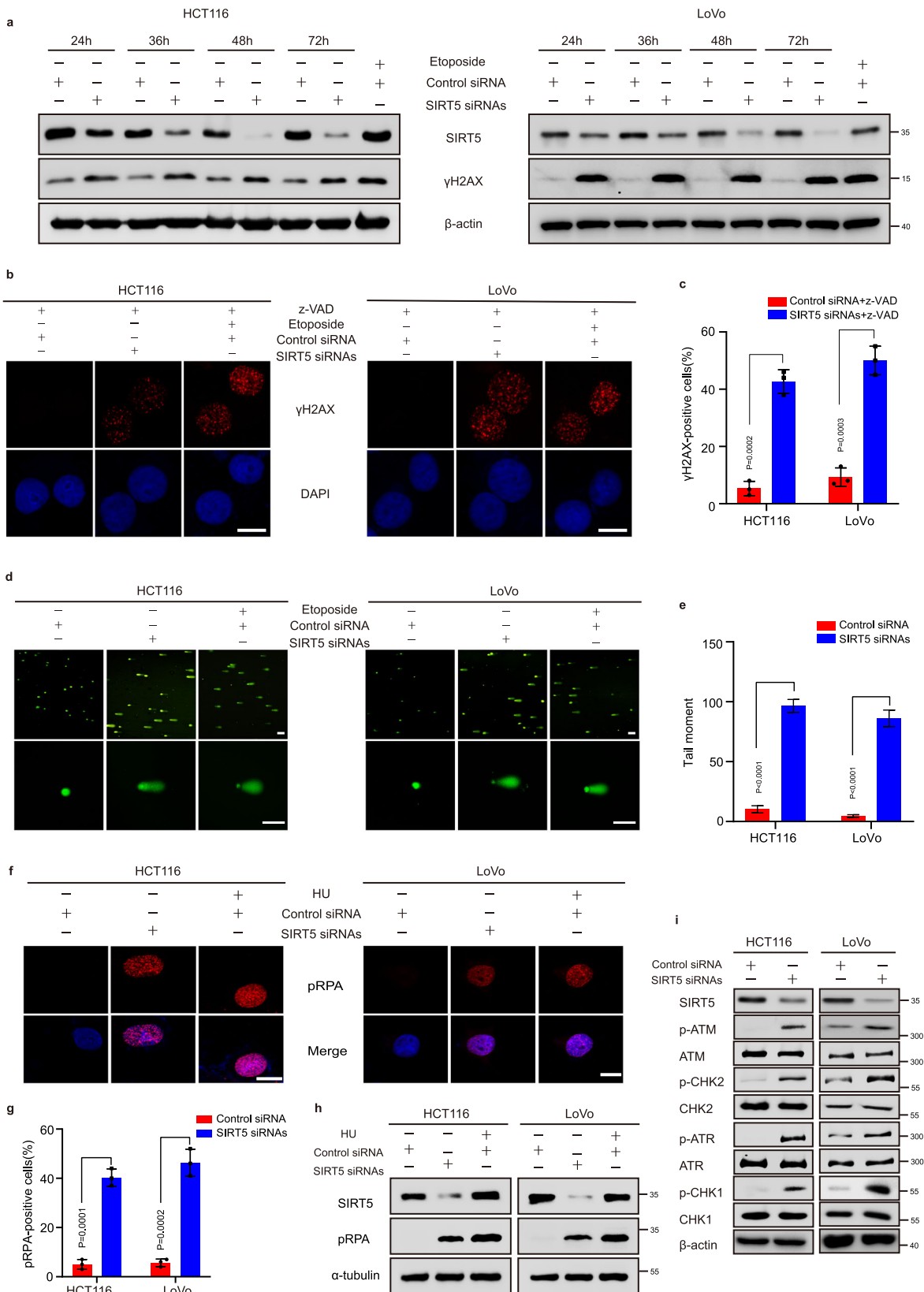

DNA oxidative damage (Supplementary Fig. 2g–i). Our findings, therefore, suggested that insufficient nucleotide synthesis, rather than ROS, was the main cause of SIRT5 silencing-induced DNA damage.

In general, cancer cells expand their nucleotide pool during the S-phase. Our previous results showed that SIRT5 inhibition reduced the levels of R5P, which contributes to nucleotide metabolism and DNA and RNA biosynthesis. We, therefore, speculated that SIRT5 suppression might considerably impact DNA synthesis in tumor cells. To validate this hypothesis, we performed the 5-ethynyl-20-deoxyuridine (EdU) assay, and found that inhibition of DNA synthesis induced by SIRT5 knockdown was reversed by supplementation with nucleosides in different CRC cell lines (HCT116, LoVo, and HT29) using

**Fig. 1 | SIRT5 silencing-induced DNA damage. a** Western blotting analysis showing the increased levels of γH2AX (Ser139) in SIRT5 siRNAs-transfected HCT116 (left) and LoVo (right) cells after 24, 36, 48, or 72 h. Exposure of CRC cell lines to 50 μM of etoposide was used as a positive control. **b, c** Immunofluorescence staining for γH2AX (Ser139) showed the increased formation of γH2AX (Ser139) nuclear foci after transfecting HCT116 and LoVo cells with SIRT5 siRNAs for 48 h. Scale bar, 5 μm. Data in (**b**) were quantified (**c**). (n = 3 biologically independent experiments). **d, e** Representative images of the alkaline comet assay in HCT116 and LoVo cells at 48 h after transfection with NC siRNA and SIRT5 siRNAs (**d**). Scale bar, 20 μm.

At least 100 nuclei were quantified for per condition (**e**). **f, g** Immunofluorescence staining for the formation of pRPA nuclear foci in HCT116 (left) and LoVo (right) cells. Scale bar, 5 μm. Data in (**f**) were quantified (**g**). (n = 3 biologically independent experiments). **h** Representative Western blots showing increased pRPA levels after SIRT5 silencing. Hydroxyurea (HU) served as a positive control. **i** Expression of p-ATM, p-CHK2, p-ATR, and p-CHK1 was elevated in CRC cells, 48 h after knocking down SIRT5. Each experiment was performed in triplicate. Values in (**c, e,** and **g**) represent mean ± standard deviation (SD). P values were calculated by a Student's t test (unpaired, two-sided). Source data are provided as a Source Data file.

immunofluorescence and flow cytometry analysis (Fig. 2i–l and Supplementary Fig. 3a–d). Besides, a similar result was obtained in HCT116 cells with stable knockdown of SIRT5 (Supplementary Fig. 3e–h).

Altogether, these results indicated that nucleotide pool deficiency was responsible for DNA damage and reduced DNA synthesis in SIRT5-silencing CRC cells.

### A low-nucleotide pool in SIRT5-knockdown CRC cells leads to cell cycle arrest and apoptosis

We examined whether the nucleotide pool influences the growth of human CRC cells. SIRT5 knockdown significantly enhanced the proportion of CRC cells in the G2/M and S phases and also increased the number of apoptotic cells, whereas exogenously supplied nucleosides decreased cell cycle arrest (Fig. 3a, b, and Supplementary Fig. 3i, j) and apoptosis (Fig. 3c, d and Supplementary Fig. 3k, l). Consistently, Western blotting revealed that the expression levels of cyclin D1 and cyclin D3, two G1 phase regulators, were downregulated, while those of cyclin E1 and cyclin A2 were upregulated in CRC cells transfected with SIRT5 siRNAs. These effects were reversed on supplementation of the four nucleosides (Fig. 3e and Supplementary Fig. 3m). In addition, the exogenous supply of nucleosides abolished SIRT5 silencing-induced upregulation of apoptosis indicators (cleaved caspase 3, caspase 8, caspase 9, PARP) (Fig. 3e and Supplementary Fig. 3m). Following this, the study performed soft agar colony formation assays and found that supplementation with four exogenous nucleosides significantly promoted anchorage-independent growth of HCT116 and LoVo cells (Fig. 3f, g). Next, on treating CRC cells with SIRT5 inhibitor 1, a newly synthesized specific human SIRT5 deacylase inhibitor, we found that DNA synthesis was impaired and cell cycle arrest and apoptosis were induced; nucleoside supplementation reversed these effects (Supplementary Fig. 4a–g). Collectively, these results suggested that SIRT5 silencing-induced DNA damage potentially resulted from a deficiency in the nucleotide pool, which is essential for supporting the extensive proliferation of CRC cells.

### SIRT5 promotes the PPP by activating TKT in a deacylation-dependent manner

Glycolytic intermediates are metabolized to R5P via the PPP, thereby supporting base ribosylation and subsequently maintaining the nucleotide pool to ensure optimal DNA replication and cell growth. Considering that R5P can be generated via two separate branches of the PPP, we used $[1,2^{-13}C_2]$-glucose as the tracer for isotopologue spectral analysis with the aim of identifying the route via which R5P is produced in LoVo cells after SIRT5 knockdown. R5P (M + 1) was produced by oxidative decarboxylation via the oxidative PPP, while R5P (M + 2) was generated via the non-oxidative PPP (Fig. 4a). It is noteworthy that SIRT5-deficient cells showed an increased level of the R5P [M + 1] isotopologue derived via $[1,2^{-13}C_2]$-glucose and a significant reduction in the level of R5P (M + 2) (Fig. 4b, c). This finding suggested that $[1,2^{-13}C_2]$-glucose was predominantly metabolized to R5P [M + 1] through the oxidative PPP, as SIRT5 knockdown blocked the non-oxidative branch.

R5P isomerase (RPI), Ru5P epimerase (RPE), TKT, and transaldolase (TALDO) are critical enzymes that regulate R5P production through the non-oxidative PPP (Fig. 4d). To determine the mechanism

via which SIRT5 drives the non-oxidized PPP to produce R5P, we assessed all the aforementioned enzymes. Western blotting revealed no significant changes in the protein levels of RPI, RPE, TKT, or TALDO (Fig. 4e). SIRT5 is a PTM enzyme that regulates the activities of various metabolic enzymes[10]. We, therefore, speculated that SIRT5 affects the activity of these enzymes and then regulates the metabolism in the non-oxidative PPP. To validate this hypothesis, immunofluorescence was performed to determine the localization of SIRT5 as well as of RPI, RPE, TKT, or TALDO. We found a strong co-localization between SIRT5 and TKT (Fig. 4f, g) but not between SIRT5 and RPI, RPE, and TALDO (Supplementary Fig. 5a–f). Further, the co-localization between TKT and SIRT5 was confirmed in CRC tissues (Supplementary Fig. 5g). These data indicated that TKT may interact with SIRT5 in CRCs. Subsequently, we studied the effect of SIRT5 on the enzymatic activity of TKT. Impressively, SIRT5 knockdown in HCT116 and LoVo cells resulted in a remarkable inhibition of TKT activity by 30% and 50%, respectively (Fig. 4h, i). Similar results were also observed in HCT116 cells with stable knockdown of SIRT5 (Supplementary Fig. 5h). Further, we found that SIRT5 inhibitor 1 inhibited TKT activity in a concentration-dependent manner (Supplementary Fig. 5i). Considering that SIRT5 is often overexpressed in CRC cells, we constructed a control vector, a SIRT5 wild-type plasmid (SIRT5 WT), and a H158Y mutant plasmid (SIRT5 H158Y, a catalytically inactive mutant without lysine deacylation activity), and reintroduced them into CRC cells. As anticipated, TKT activity was significantly elevated in cells overexpressing SIRT5 WT, while there was little change in those overexpressing the mutant plasmid and control vector (Fig. 4j).

Subsequently, to investigate whether the effect of SIRT5 on nucleotide pool is related to its deacylation activity, we used targeted metabolomics to assess nucleotide levels in the cell lines stably expressing the control vector, SIRT5 WT, and SIRT5 H158Y. We found that in comparison with the control vector and SIRT5 H158Y groups, nucleotide levels in cells overexpressing SIRT5 were significantly increased (Fig. 4k), indicating that SIRT5 promotes the PPP by activating TKT in a deacylation-dependent manner.

### TKT contributes to SIRT5 knockdown-induced nucleotide pool deficiency and DNA damage in CRC cells

TKT, as a key enzyme in the non-oxidative PPP, supplies more R5P to facilitate tumor proliferation. Herein we investigated whether TKT contributes to the production of precursors required for SIRT5-mediated nucleotide biosynthesis, and found that TKT knockdown abrogated SIRT5-induced increase in nucleotides in HCT116 cells (Fig. 5a). In addition, EdU assay results revealed that TKT overexpression reversed the decrease in DNA synthesis caused by SIRT5 knockdown (Fig. 5b, c). Moreover, in comparison with the control vector, DNA damage in cells overexpressing SIRT5 was significantly decreased upon fluorouracil (5-FU) treatment, a DNA damaging agent, which was reversed by the knockdown of TKT (Fig. 5d). Consistently, treatment with TKT inhibitor oxythiamine (OT) abolished SIRT5-induced protection from DNA damage (Fig. 5e). Flow cytometry analysis of cell cycle and apoptosis showed that TKT overexpression rescued the effects observed on SIRT5 silencing (Fig. 5f–i). In addition, Western

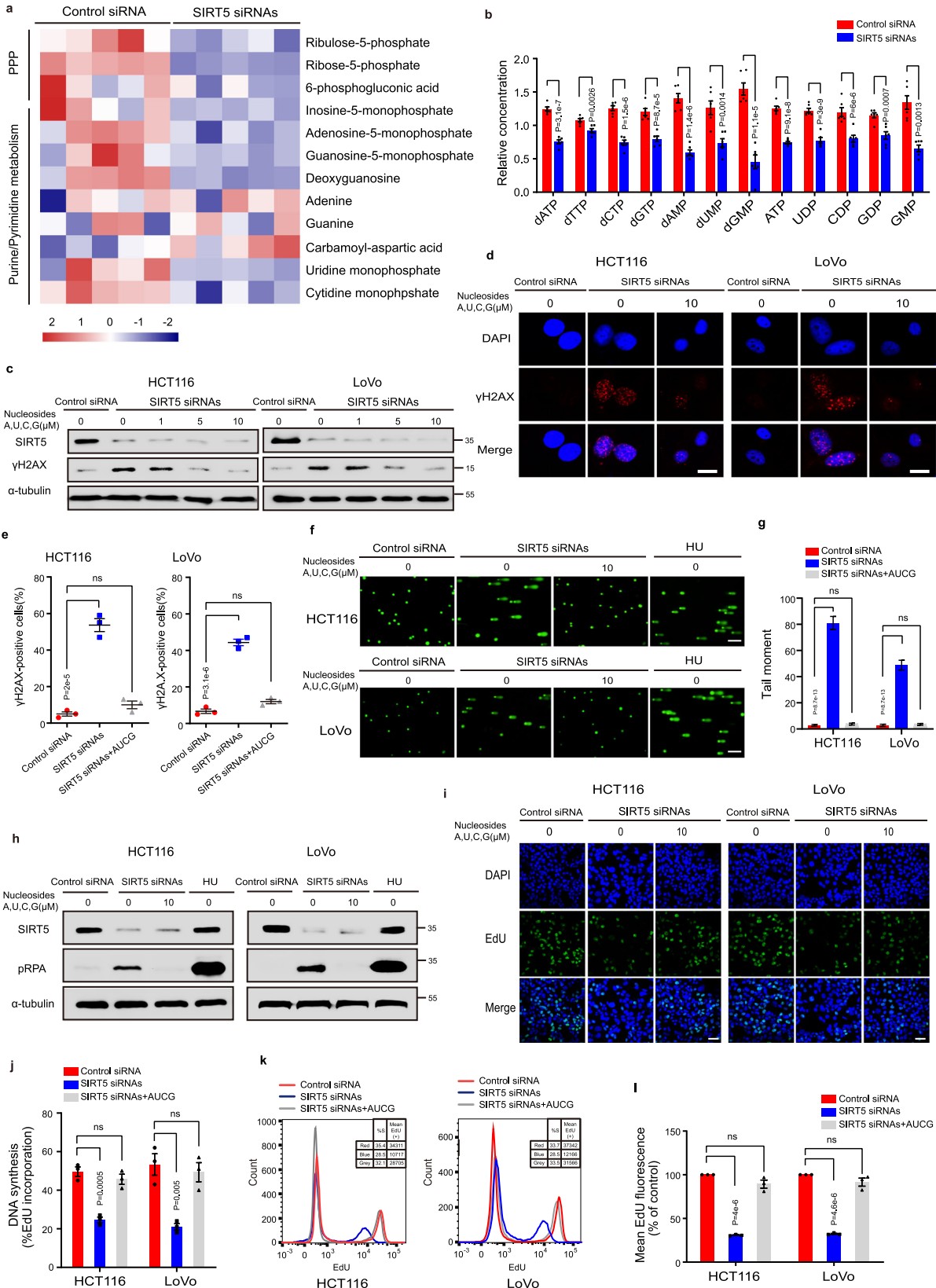

blotting validated that TKT overexpression blocked the upregulation of apoptotic pathway proteins and γH2AX induced by SIRT5 silencing (Fig. 5j). Collectively, these results suggested that SIRT5 knockdown resulted in nucleotide pool deficiency in a TKT-dependent manner, resulting in impaired DNA synthesis and DNA damage in CRC cells.

## SIRT5 activates TKT by mediating its demalonylation

Considering that a strong co-localization was observed between SIRT5 and TKT in HCT116 and LoVo cells (Fig. 4f, g), we explored whether the deacylation activity of SIRT5 had an impact on their localization. Endogenously expressed TKT and FLAG-SIRT5 showed a strong co-localization (Fig. 6a, b), which was independent of the catalytic activity

**Fig. 2 | SIRT5 sustains the nucleotide pool by enhancing R5P synthesis to maintain DNA stability. a** Heatmap showing significantly differentially expressed metabolites in the PPP and purine/pyrimidine metabolism pathway after SIRT5 deletion in HCT116 cells. ($n$ = 5 biologically independent experiments). **b** Targeted metabolomics analysis of nucleotides in HCT116 cells after SIRT5 knockdown. Data are expressed as mean ± standard error of the mean (SEM). Statistical significance was determined using the two-sided Student's $t$ test. ($n$ = 6 biologically independent experiments). **c–e** Exogenous supplementation with nucleosides decreased the levels of γH2AX induced by SIRT5 silencing. Cells were transfected with SIRT5 siRNAs for 48 h and then cultured at indicated concentrations with four nucleosides (A, U, C, and G) for 16 h. Immunoblotting (**c**) and immunofluorescence staining (**d**) for γH2AX. Scale bars, 5 μm. Quantitation of the percentage of γ-H2AX-positive cells (**e**). ($n$ = 3 biologically independent experiments). **f, g** Representative images of the alkaline comet assay for the SIRT5-knockdown HCT116 and LoVo cells under the indicated condition. Cells were cultured with four nucleosides (10 μM) for 16 h (**f**). Scale bars, 20 μm. Data in (**f**) were quantified (**g**). At least 100 nuclei were quantified for per condition (**e**). **h** Western blot showed that the pRPA levels were decreased in SIRT5-silenced CRC cells after supply of exogenous nucleosides. HU was used as a positive control. **i–l** Exogenous supplementation with the four nucleosides restored DNA synthesis in CRC cells after SIRT5 knockdown. EdU assay involving immunofluorescent staining (**i**) and flow cytometry (**k**). Scale bars, 20 μm. The data in (**i** and **k**) were quantified and analyzed in (**j**) and (**l**) respectively. ($n$ = 3 biologically independent experiments). Data are presented as mean values ± SEM. Statistical significance was calculated using one-way ANOVA corrected with Tukey's multiple comparisons test. ns not significant. Source data are provided as a Source Data file.

of SIRT5 as SIRT5 mutants did not show impaired binding with TKT. The interaction between SIRT5 and TKT was further validated using the co-immunoprecipitation assay (Fig. 6c, d).

As SIRT5 activated TKT in a deacylation-dependent manner, we hypothesized that the direct interaction between TKT and SIRT5 would favor the induction of lysine deacylation of TKT. In a recent proteomic study[2], mouse TKT was detectably demalonylated by SIRT5 at six lysine residues. Therefore, we detected lysine malonylation levels of TKT upon SIRT5 treatment. The results revealed significantly diminished lysine malonylation levels of TKT in cells overexpressing SIRT5 but not the mutant (Fig. 6e). In contrast, lysine malonylation levels of TKT were elevated in SIRT5-silenced HCT116 and LoVo cells (Fig. 6f). SIRT5 is reportedly localized in both the mitochondria and cytoplasm. We observed that TKT was distributed in the cytoplasm as well as the mitochondria of HCT116 cells using subcellular fractionation and immunoelectron microscopy (Supplementary Fig. 5j, k). Similarly, we found that TKT may be demalonylated by SIRT5 in both mitochondrial and cytoplasmic fractions (Supplementary Fig. 5l). Although we validated that SIRT5 affected the lysine malonylation level of TKT, it was unclear whether TKT malonylation affected its enzymatic activity. As acyl-CoA can serve as the donor molecule for lysine acylation modification[8], immunoprecipitated hemagglutinin (HA)-tagged TKT was incubated with malonyl-CoA and TKT activity was then measured. We found a prominent decrease in TKT activity, suggesting that lysine malonylation of TKT inhibited its activity (Fig. 6g, h).

Among the six malonylated sites on TKT, the malonylation level of lysine-281 showed a considerable change in the absence of SIRT5[2]. Lysine-281 is conserved in TKT orthologs from humans to *Gallus gallus*, suggesting that it is critical to the function of TKT (Fig. 6i). As previously reported[33], malonylation alters the charge of lysines by changing the positive charge of the ε-amino group to a negatively charged carboxylic acid. To assess whether the modification of lysine-281 can affect the enzymatic activity of TKT, we constructed three mutant plasmids, substituting lysine (K) 281, K282, and K283 with arginine (R) 281, R282, and R283, respectively, with the latter retaining a positive charge. Subsequently, HCT116 cells ectopically expressing WT TKT and K281R, K282R, and K283R mutants were treated with SIRT5 siRNAs, and Western blotting was performed to analyze the malonylation level of TKT. The K281R mutation caused a significant reduction in the malonylation level of TKT (Fig. 6j). Moreover, suppressing SIRT5 increased the malonylation level of TKT in HCT116 cells expressing WT TKT as well as in K282R and K283R mutants, but not in K281R mutant, suggesting that TKT was demalonylated in a SIRT5-dependent manner on lysine 281 (Fig. 6j). Further, the K281R mutant did not respond to SIRT5-mediated regulation of TKT activity (Fig. 6k), indicating that lysine-281 in TKT is a major malonylation site of SIRT5. Altogether, these results suggested that SIRT5 elevates the activity of TKT via demalonylation.

## Impact of TKT on SIRT5-mediated tumorigenesis in vivo

To study the role of the SIRT5–TKT axis on tumor behavior in vivo, we generated a surgical orthotopic mouse model by injecting luciferase-transfected HCT116 cells stably expressing NTC shRNA or SIRT5 shRNAs into the cecum of nude mice. Bioluminescence imaging results are shown in Fig. 7a, b. We found that in comparison with the control group, both tumor volume and weight in the SIRT5-silencing group were significantly restrained (Fig. 7c–e). Besides, relative to the control group, TKT activity was decreased by 38% in SIRT5-knockdown tumors (Fig. 7f). Consistent with our in vitro results, targeted metabolomics analysis of tumor lysates revealed that SIRT5 silencing resulted in a significant downregulation in the levels of R5P and nucleotides (Fig. 7g). Western blotting was performed using the lysates of orthotopic tumors, confirming the stable knockdown of SIRT5 (Fig. 7h). As anticipated, SIRT5 silencing considerably increased γH2AX levels in orthotopic tumors (Fig. 7h). These findings were further verified in the subcutaneous xenograft tumor model. In support of the SIRT5–TKT axis, TKT overexpression rescued SIRT5 silencing-induced decreased tumor volume and weight, downregulation of R5P and nucleotide levels, as well as DNA damage and cell apoptosis. (Fig. 7i–o). Considering that SIRT5 is often overexpressed in CRCs, we established a subcutaneous xenograft tumor model in nude mice by injecting them with HCT116 cells stably expressing the control vector, SIRT5 WT, and SIRT5 H158Y. The overexpression of SIRT5 WT was found to markedly accelerate CRC tumorigenesis (Supplementary Fig. 6a–c). Moreover, in comparison to control tumors, TKT activity was increased by 51% in SIRT5 WT tumors, while it remained similar between the SIRT5 H158Y and control groups (Supplementary Fig. 6d). Targeted metabolomics analysis of tumor lysates revealed that SIRT5 overexpression increased the amount of R5P and nucleotides (Supplementary Fig. 6e). Collectively, these results supported our in vitro data, validating that SIRT5 plays a key role in CRC tumorigenesis by activating TKT, and sufficient nucleotide levels are consequently maintained for DNA synthesis.

## SIRT5 levels are correlated with γH2AX levels and could predict CRC patient outcome

To further address the clinical significance of SIRT5-mediated DNA damage in CRC carcinogenesis, we explored the correlation between SIRT5 and γH2AX expression in 60 human CRC specimens. Immunohistochemistry (IHC) revealed that SIRT5 expression was negatively correlated with γH2AX levels ($p < 0.001$, Fig. 8a, b). Based on the fact that SIRT5 preserves nucleotide pool and protects cells from DNA damage, we hypothesized that high-SIRT5 levels could be associated with chemoresistance. To verify this speculation, we detected the sensitivity of HCT116 and LoVo cells stably expressing the control vector, SIRT5 WT, and SIRT5 H158Y to 5-FU. In comparison with the control vector and SIRT5 H158Y groups, cell inhibition rate analysis showed that cancer cells overexpressing SIRT5 WT demonstrated a significantly higher survival rate on 5-FU treatment (Fig. 8c). Further, in the SIRT5 WT group, 5-FU-induced apoptosis of HCT116 and LoVo cells was reduced using flow cytometry (Fig. 8d), which was confirmed by

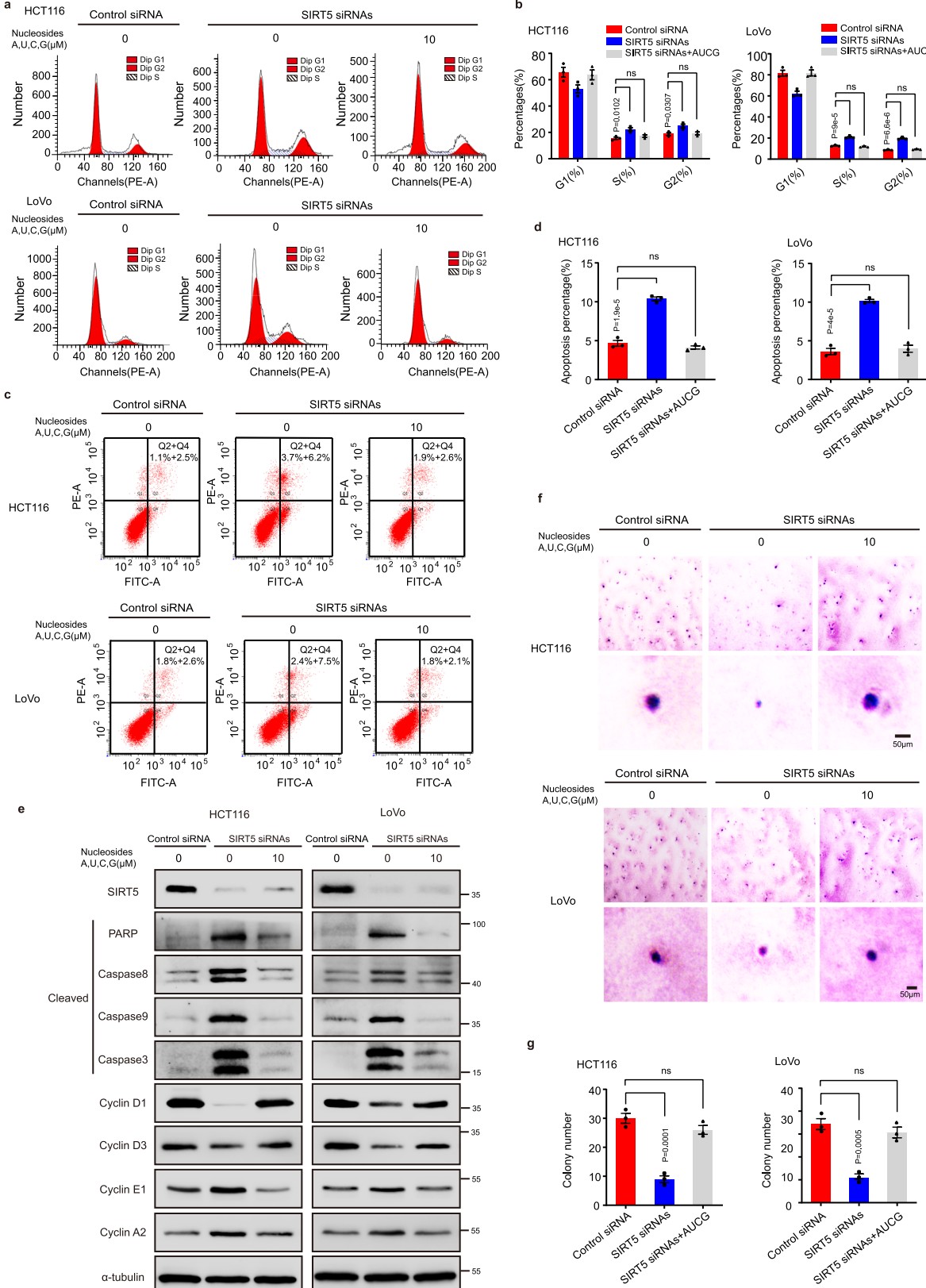

**Fig. 3 | Exogenous nucleoside supplementation reduces the effects of SIRT5 silencing on cell cycle, apoptosis, and colony formation in CRC cells. a–d** The flow cytometry was used to detect changes in the cell cycle (**a**) and apoptosis (**c**) in SIRT5-deficient HCT116 and LoVo cells with exogenous nucleoside supplementation for 16 h. The data in (**a** and **c**) were quantified and analyzed in (**b**) and (**d**) respectively. **e** The expression of apoptosis indicators (the cleaved caspase 8, caspase 9, caspase 3, and PARP) and cell cycle regulators in SIRT5-silenced CRC cells after exogenous supplementation with nucleosides were detected by Western blot. **f, g** Representative images (**f**) and quantification (**g**) using soft agar colony formation assay in SIRT5-silenced CRC cells with or without four nucleosides. Scale bar, 50 µm. Values in (**b**, **d**, and **g**) represent mean ± SEM from three independent experiments. One-way ANOVA with Tukey's multiple comparisons test was used. ns, not significant. Source data are provided as a Source Data file.

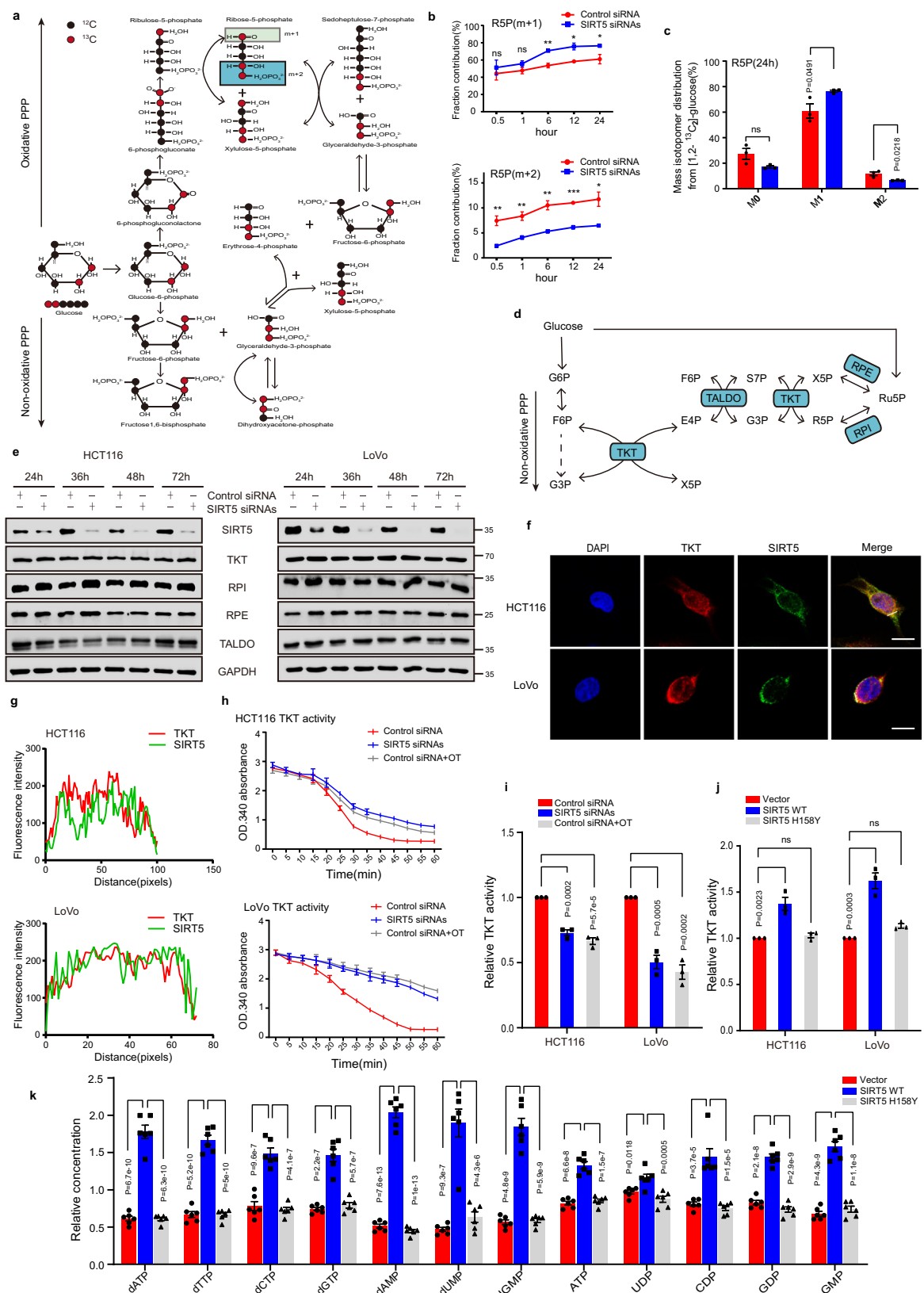

Western blotting (Fig. 8e). In the CRC xenograft mouse models, HCT116 cells ($2 \times 10^6$ cells) stably expressing the control vector, SIRT5 WT, and SIRT5 H158Y were subcutaneously injected. Interestingly, 5-FU treatment significantly decreased tumor growth and weight in the control vector and SIRT5 H158Y groups, but this effect was not notable in the SIRT5 WT group (Fig. 8f–h). Collectively, these results indicated

that SIRT5 WT is associated with chemoresistance to 5-FU. To decipher the clinical relevance of SIRT5 expression with chemoresistance, we evaluated the relationship between SIRT5 expression and overall survival in the Gene Expression Omnibus dataset (raw data accessible via GSE72970) in which CRC patients were treated with FOLFOX or FOL-FIRI. We found that the high levels of SIRT5 in CRC tissues were

**Fig. 4 | SIRT5 promotes the non-oxidative PPP by activating TKT. a** Schematic model of the PPP metabolism in cancer cells. Red circles represent carbons derived from [1,2-$^{13}$C$_2$] glucose, and black circles are the unlabeled. [1,2-$^{13}$C$_2$]glucose is converted to R5P (M + 1) through the oxidative PPP and R5P (M + 2) is generated from the non-oxidative PPP. **b, c** Ratio of R5P (M + 1) to R5P (M + 2) from [1,2-$^{13}$C$_2$]-glucose was determined after SIRT5 knockdown; cells were transfected with SIRT5 siRNAs for 48 h and then cultured with fresh medium containing [1,2-$^{13}$C$_2$]-glucose (11.1 mM) for indicated time points (**b**). Quantitative analysis of the R5P sources at 24 h (**c**). Metabolite levels were normalized to the cell number. (*n* = 3 biologically independent experiments). Values in (**b** and **c**) represent means ± SEM. Statistical significance was calculated using a two-tailed unpaired *t*-test. **d** Schematic model of key enzymes involved in the non-oxidative PPP. **e** Expression of TKT, RPI, RPE, and TALDO after SIRT5 depletion. **f, g** Immunofluorescent staining of SIRT5 (in green) and TKT (in red); yellow in the merged magnified images

indicates their co-localization (**f**). Scale bar, 5 μm. The fluorescence intensity of SIRT5 (green line) and TKT (red line) was traced along the white line in CRC cells using the line profiling function of ImageJ (**g**). **h, i** TKT activity was determined following SIRT5 knockdown in HCT116 and LoVo cells. Representative images (**h**) and quantification of TKT activity (**i**). (*n* = 3 biologically independent experiments). The TKT inhibitor oxythiamine (OT; 20 μM) served as a positive control. **j** TKT activity was determined in HCT116 and LoVo cells stably expressing the control vector, SIRT5 WT, or SIRT5 H158Y. Quantification of TKT activity. (*n* = 3 biologically independent experiments). **k** Targeted metabolomics analysis of nucleotide levels in HCT116 cells stably expressing the control vector, SIRT5 WT, or SIRT5 H158Y. (*n* = 6 biologically independent experiments). Values in (**h**–**k**) represent mean ± SEM. One-way ANOVA with Tukey's multiple comparison test was performed. *\*p* < 0.05, *\*\*p* < 0.01, *\*\*\*p* < 0.001. ns, not significant. Source data are provided as a Source Data file.

## Discussion

SIRT5 has been widely reported to be overexpressed in various types of cancer, including CRC[24], hepatocellular carcinoma[34], and ovarian cancer[14], and it is involved in regulating tumor survival and progression. Herein our data indicated that SIRT5 silencing increased the lysine malonylation levels of TKT, thereby suppressing the non-oxidative PPP and leading to insufficient R5P levels for nucleotide synthesis, which contributed to DNA damage and growth inhibition of tumor cells (Fig. 9).

SIRTs, including SIRT1[35,36], SIRT2[37], SIRT3[38], SIRT4[39], SIRT6[40,41], and SIRT7[42,43], maintain genomic integrity by directly deacetylating components of the DNA repair machinery or indirectly decreasing the production of ROS via metabolic reprogramming. However, the role of SIRT5 in regulating DNA damage remains largely unclear. In the present study, we provide evidence that SIRT5 knockdown causes DNA damage in CRC cells. DDR is widely known to be a complex and orderly mechanism that is induced in response to DNA damage[44]. Although DDR systems, including the ATM-CHK2 and ATR-CHK1 pathways, were activated following SIRT5 silencing, DNA damage persisted.

Compelling evidence suggests that nucleotide deficiency induces DNA damage due to stalling of the replication fork and production of mismatched DNA[30,45]. Thus, we speculated that SIRT5 causes DNA damage by regulating the supply of nucleotides. Our results indicated that SIRT5 knockdown reduced R5P and nucleotide levels in CRC cells; moreover, SIRT5 silencing-induced DNA damage was reversed by the exogenous supplementation of the four nucleosides (A, U, C, and G). As previously reported, unrepaired DNA damage can cause cell apoptosis via the caspase-dependent apoptosis pathway[46]. We have shown here that exogenous nucleoside supplementation inhibited the caspase-dependent apoptosis pathway, which was elicited by SIRT5 silencing. Our results thus demonstrated that the role of SIRT5 in DNA damage was mainly exerted by maintaining the intracellular nucleotide pool, finally affecting the cell cycle and apoptosis. We further found that SIRT5 could be a potential anticancer target, whose silencing induced DNA damage and rendered CRC cells more sensitive to 5-FU.

To meet the high nucleotide demand of tumor cells, the PPP becomes peculiarly active[47]. A previous study revealed that ~80% of the R5P required for nucleotide synthesis is provided by the non-oxidative PPP in cancer cells[20], while normal cells generate R5P via the oxidative PPP. Our findings demonstrated that SIRT5 silencing specifically inhibited the non-oxidative PPP, indicating that SIRT5 might be a promising therapeutic target. TKT is the key enzyme in the non-oxidative PPP and controls the ratio of the oxidative PPP versus non-oxidative PPP flux. Cancer cells evidently increase the metabolic flux of the non-oxidative PPP for R5P production via TKT under hypoxia[48]. Another study reported that active TKT maintains the non-oxidative

PPP flux to generate R5P for nucleic acid synthesis[49]. Herein we observed that SIRT5 increased the non-oxidative PPP flux by activating TKT in a PTM. However, this might not be the only mechanism to upregulate or activate TKT. TKT expression is also reportedly upregulated at the transcriptional level to promote the non-oxidative PPP[23]. Furthermore, our results showed that TKT is crucial for SIRT5-mediated carcinogenesis not only in vitro but also in vivo. In addition, it has been reported that TKT counteracts oxidative stress by supplementing NADPH, which is beneficial to cancer growth[50]. This may partly explain why SIRT5 silencing increased ROS levels in CRC cells.

The reversible malonylation of lysines is involved in numerous metabolic processes, including glycolysis, gluconeogenesis, urea cycle, and fatty acid β-oxidation[2]. However, it remains unclear how lysine demalonylation affects the PPP. The results of this study revealed an interaction between SIRT5 and TKT, which led to K281 demalonylation and subsequent activation of TKT in CRC. A previous study reported that TKT phosphorylation affects its enzymatic activity in human cervical cancer cells[51]. However, due to the negatively charged nature and large size, malonylation is more likely to have a profound effect on protein structure and function in comparison to other modifications[2,52]. Collectively, these findings provide deeper insights into the role of malonylation in regulating the PPP, which is critical for DNA integrity and cell survival. Nonetheless, further studies are warranted to elucidate specific mechanisms.

In conclusion, we herein report that SIRT5 regulates the non-oxidative PPP by activating TKT in a demalonylation-dependent manner, consequently increasing R5P generation and supporting nucleotide synthesis. This in turn affects DNA damage and cell proliferation in CRC. Collectively, our data provide a better understanding of the close interaction among SIRT5, cell metabolism, and DNA damage, and also suggest that SIRT5 can serve as a promising target for CRC treatment.

## Methods
### In vivo models
Five-week-old male BALB/c nude mice were used for all xenograft experiments. The animals were weighed and randomly divided into different groups. The orthotopic CRC mouse model was established as previously described[53]. Stable HCT116 cells expressing either NTC shRNA-luciferase (Luc) or SIRT5 shRNAs-Luc were generated. Mice were anesthetized in an acrylic chamber with 2.5% isofluorane/air mixture; $1 \times 10^5$ HCT116 cells (HCT116-NTC shRNA-Luc or HCT116-SIRT5 shRNAs-Luc) were suspended in 30 μL PBS medium and injected into the cecal wall of 10 5-week-old male nude mice. To prevent leakage, a cotton swab was cautiously held for 1 min over the injection site. Tumor volumes were weekly monitored using a bioluminescence imaging system (IVIS® Lumina Series III) and the IVIS Lumina Series III Software (version 4.7.4, PerkinElmer). Before imaging, the animals were anesthetized in an acrylic chamber with 2.5% isofluorane/air mixture and intraperitoneally injected with D-luciferin potassium salt

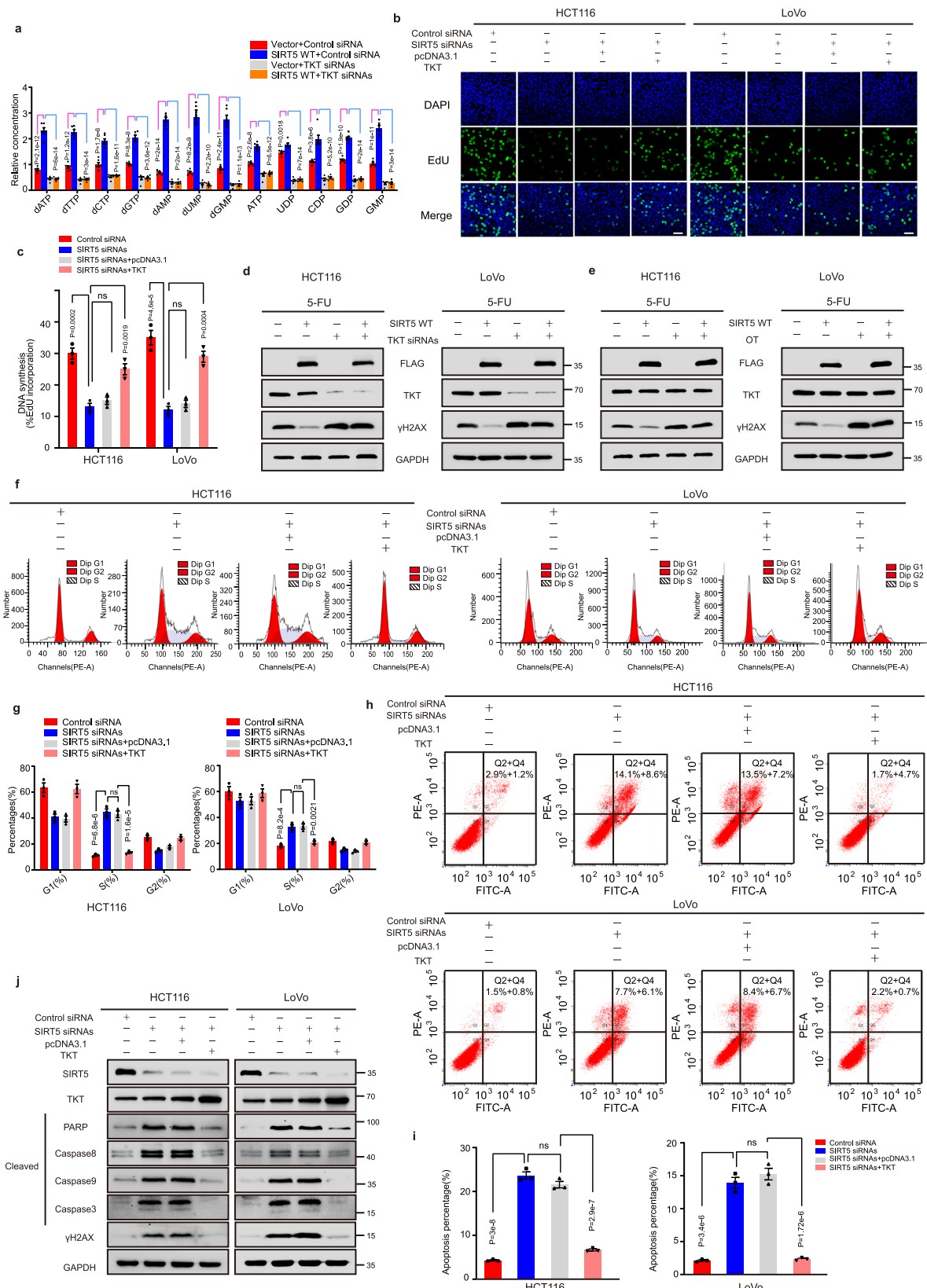

in PBS at a dose of 15 mg/kg body weight. After incubation for 10 min with luciferin, a digital grayscale image was acquired, followed by the acquisition and overlay of a pseudocolor image representing the spatial distribution of detected photons emerging from active luciferase within the animal. Signal intensity was quantified as the sum of all detected photons within the region of interest per second per

steradian. The animals were imaged on day 0, 7, 14, 21, and 28 of treatment. Primary tumors in the cecum were excised, and the final tumor volume was measured as (shortest diameter)² × (longest diameter) × 0.5.

For the rescue function experiments, nude mice were subcutaneously injected with 2 × 10⁶ HCT116 cells. Thereafter, the animals

**Fig. 5 | TKT protects CRC cells from DNA damage, cell cycle arrest, and apoptosis after SIRT5 knockdown by the maintaining nucleotide pool. a** Targeted metabolomics analysis of nucleotide levels in HCT116 cells stably expressing the control vector or SIRT5 WT treated with a control siRNA or siRNAs targeting TKT for 48 h. ($n = 6$ biologically independent experiments). **b, c** Representative immunofluorescence images of the EdU incorporation assay were captured in SIRT5-deficient HCT 116 and LoVo cells, after transfection with an empty vector or TKT plasmid (**b**). Scale bar, 5 μm. Data in (**b**) were quantified (**c**). ($n = 3$ biologically independent experiments). **d, e** Immunoblotting of γ-H2AX in HCT116 and LoVo cells stably expressing the control vector and SIRT5 WT, followed by treatment with TKT siRNAs (**d**) or OT (20 μM, **e**). 20 μM 5-FU was used as a DNA-damaging

agent. **f–i** Flow cytometry was used to detect the effect of overexpressing TKT on the cell cycle (**f**) and apoptosis (**h**) in SIRT5-silenced CRC cells. The data in (**f** and **h**) were quantified and analyzed in (**g**) and (**i**) respectively. ($n = 3$ biologically independent experiments). **j** Western blotting showing that TKT overexpression inhibited the increased levels of cleaved caspase 3, caspase 8, caspase 9, PARP, and γH2AX in SIRT5-silenced HCT116 and LoVo cells. Values in (**a, c, g,** and **i**) represent mean ± SEM. Experiments in (**b–i**) were performed three times independently with similar results. One-way ANOVA with Tukey's multiple comparisons test was used for assessing significance. ns, not significant. Source data are provided as a Source Data file.

were randomly divided into four groups, including the control vector or TKT WT groups with or without SIRT5 knockdown. Two adenoviruses targeting the SIRT5 and TKT genes were then administered every 3 days. Tumor length and width (in millimeters) were measured every 3 days using calipers, and tumor volume was calculated using this formula: (shortest diameter)$^2$ × (longest diameter) × 0.5. Tumors were eventually dissected and analyzed.

To explore the role of SIRT5 in CRC chemoresistance in vivo, $2 \times 10^6$ HCT116 cells stably expressing the control vector, SIRT5 WT, and SIRT5 H158Y were subcutaneously injected into each mouse to establish the CRC xenograft model. Six days later, 5-FU (30 mg/kg) was administered via intraperitoneal injection every 3 days. We designed six groups: (i) control vector and saline, (ii) SIRT5 WT overexpression and saline, (iii) SIRT5 H158Y overexpression and saline, (iv) control vector and 5-FU, (v) SIRT5 WT overexpression and 5-FU, and (vi) SIRT5 H158Y overexpression and 5-FU groups. All mice were housed under pathogen-free with a maximum of five mice per cage. Mice were maintained in a standard environment with a relative humidity of 50% and 12 h light/dark cycle at 25 °C.

The mice were euthanized when the tumor exceeded 10% of the mouse body weight. All animal studies were conducted according to the guidelines approved by the Institutional Animal Care and Use Committee of Renji Hospital, Shanghai Jiao tong University School of Medicine. The approval number for animal experiments is 2019-0024.

## Cell culture
The human CRC cell lines HCT116 and LoVo were obtained from the American Type Culture Collection (Manassas, VA, USA) and maintained at 37 °C in a humidified incubator (5% CO2) in McCoy's 5 A (Gibco BRL, Grand Island, NY, USA) supplemented with 10% fetal bovine serum (Gibco BRL) and penicillin-streptomycin. The cell lines were free of *Mycoplasma* via MycoBlue Mycoplasma Detector (Vazyme Biotech, Nanjing, China). The following chemicals were also added to the culture media: Camptothecin (catalog #CSN16581), 5-FU (#CSN19496), and Z-VAD-FMK (#CSN19230). All of them were purchased from CSNpharm (Chicago, USA). Adenosine (catalog #58-61-7), guanosine (#118-00-3), cytidine (#65-46-3), and uridine (#58-96-8) were obtained from Sangon Biotech (Shanghai, China). SIRT5 inhibitor 1 (catalog #2166487-21-2) and the TKT inhibitor OT (#136-16-3) were from MedChemExpress (MCE, USA).

## siRNA transfection
siRNAs specifically targeting SIRT5 have been previously described. SIRT5 siRNAs were composed of the following sequences: siRNA-1, 5′- GCUGGAGGUUAUUGGAGAAUT-3′ and siRNA-2, 5′-GUGGCUG AGAAUUACAAGAUT-3′. Further, the siRNA specifically targeting TKT was purchased from GenePharma (Shanghai, China). The TKT siRNAs were composed of 5′-CCAGCCAACAGCCAUCAUUTT-3′ and 5′-CCGGC AAAUACUUCGACAAUT-3′. These siRNAs were transfected into subconfluent cells using the DharmaFECT-1 transfection reagent (Dharmacon, Lafayette, CO, USA), according to manufacturer instructions. Each transfection reaction was performed in six-well plates. Briefly,

30% confluent CRC cells were transfected with 1 μg siRNA and 5 μL DharmaFECT-1 transfection reagent in 100 μL Opti-MEM medium (Invitrogen). A nonspecific siRNA served as the negative control (NC siRNA).

## Western blotting
Cells or tissues were collected and lysed in radioimmunoprecipitation assay lysis buffer supplemented with a protease inhibitor cocktail (Kangcheng, Shanghai, China). Proteins were separated via SDS-polyacrylamide gel electrophoresis and then immunoblotted. The following primary antibodies were used: anti-SIRT5 (catalog #HPA022002, 1:2000, Sigma-Aldrich), anti-γH2AX (#9718, 1:1000), anti-p-ATM (#5883, 1:1000), anti-p-ATR (#2853, 1:1000), anti-p-CHK1 (#2348, 1:1000), anti-p-CHK2 (#2197, 1:1000), anti-cleaved caspase 3 (#9664, 1:1000), anti-cleaved caspase 8 (#9496, 1:1000), anti-cleaved caspase 9 (#7237, 1:1000), anti-cleaved PARP (#5625, 1:1000), anti-cyclin D1 (#2978, 1:1000), anti-cyclin D3 (#2936, 1:1000), anti-cyclin A2 (#4656, 1:1000), anti-cyclin E1 (#20808, 1:1000), anti-COX IV (3E11) (#4850, 1:1000), anti-α-tubulin (#2148, 1:2000), anti-p-p53 (Ser15) (#9286, 1:1000), (all from Cell Signaling Technology, Danvers, MA, USA), anti-ATM (#70103, 1:1000), anti-ATR (#70109, 1:1000), anti-TKT (#101477, 1:1000), and anti-TALDO1 (#102076, 1:1000) (all from GeneTex, California, TX, USA). Besides, anti-CHK1 (catalog #8048, 1:1000), anti-CHK2 (#5278, 1:1000), anti-RPI (#515328, 1:1000), and anti-RPE (#393655, 1:1000) were purchased from Santa Cruz Biotechnology (Santa Cruz, CA, USA) and anti-FLAG (#F1804, 1:1000) were from Sigma-Aldrich. Anti-hemagglutinin (#MMS-101P, 1:1000) was obtained from Convance (Princeton, NJ, USA) and anti-pan malonylation (#PTM-901, 1:1000) was from PTM Biolabs (Hangzhou, China). anti-phospho RPA32 (S4/S8) (catalog #A300-245A, 1:1000) was purchased from Bethyl Laboratories (Montgomery, TX, USA). GAPDH-HRP (catalog #KC-5G5, 1:5000, Kangcheng, China). Anti-beta actin (catalog #KC-5A08, 1:2000, Kangcheng, China). The secondary antibodies included peroxidase-conjugated anti-rabbit and anti-mouse (1:5000, Kangcheng) antibodies. Western blotting bands were detected using an ECL Western blotting substrate (Thermo Scientific, Waltham, MA, USA) and scanned using ChemiDoc™ MP Imaging System (Bio-Rad). Immunoblots were quantified by Image J software (Image J, version 2.1.0).

## Assessment of cell cycle and apoptosis
Flow cytometry was performed to analyze the progression of the cell cycle and apoptosis. Briefly, HCT116 and LoVo cells were cultured under the aforementioned conditions. To assess the cell cycle, cells were stained with 50 μg/mL propidium iodide (PI) containing 20 μg/mL DNase-free RNase, after which they were analyzed using flow cytometry, according to manufacturer instructions. The G1, S (DNA synthesis), G2, and M (mitosis) phases were then identified based on DNA content and the percentage of cells was determined in the distinct phases. All experiments were performed at least three times. The same analysis was also performed by double staining cells with fluorescein isothiocyanate (FITC)-conjugated Annexin V (Annexin V-FITC) and PI (BD Pharmingen, San Diego, CA, USA), followed by flow cytometry.

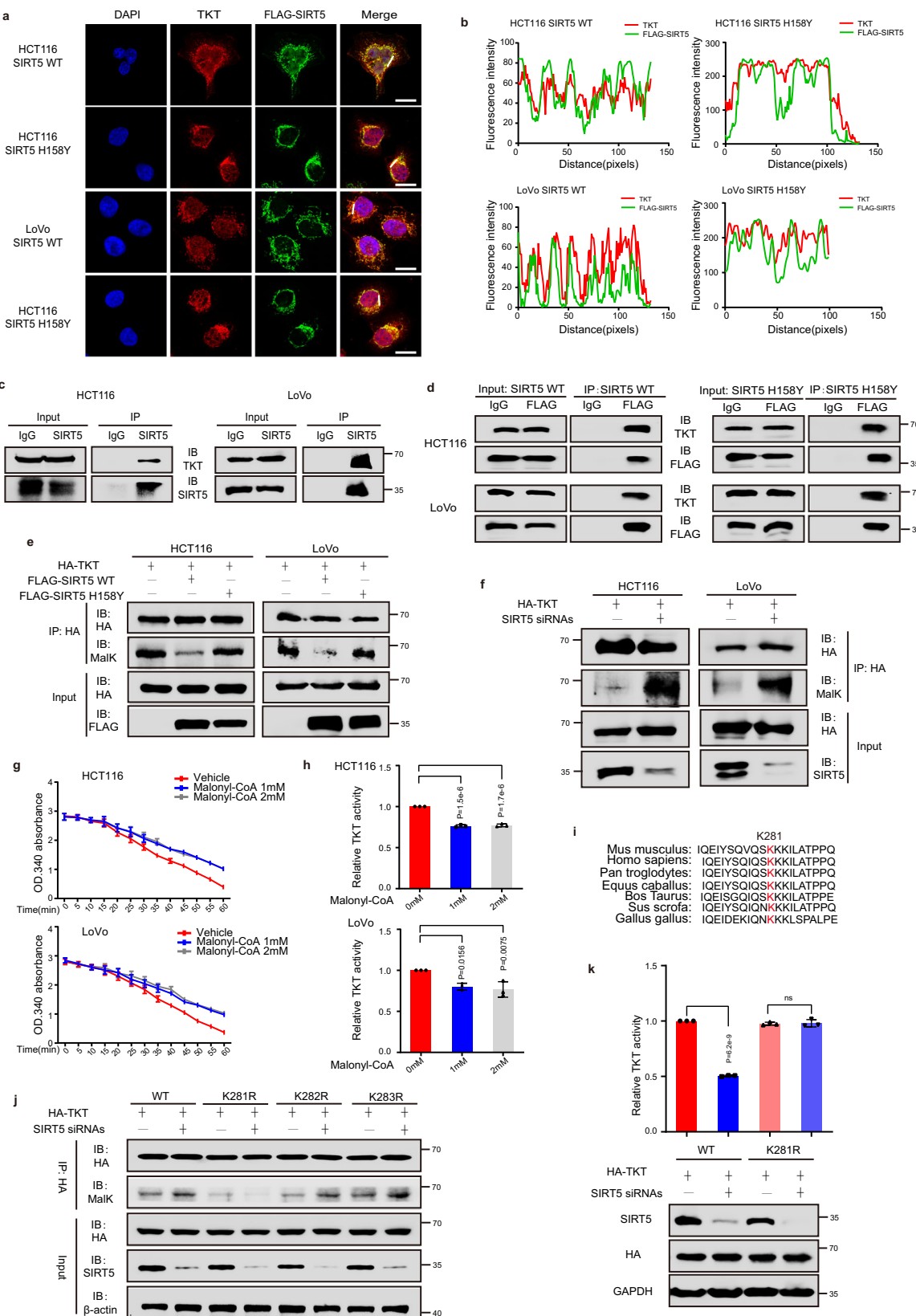

## EdU uptake

Half of the medium was replaced with fresh media containing 20 μM EdU for 2 h, followed by fixation in 3.7% formaldehyde in PBS. After 0.5% Triton X-100 permeabilization, 500 μL Click-iT® reaction cocktail was added per coverslip, according to manufacturer instructions (Click-iT EdU Image Kit, Life Technologies, catalog

#C10338). Thereafter, the Hoechst 33342 dye was bound to DNA, and the percentage of EdU⁺ cells was quantified using the Zeiss digital image processing software, ZEN® (blue edition). In the EdU flow cytometry assay, EdU-labeled cells were visualized by the Click-iT EdU Pacific Blue Flow Cytometry Assay kit (Life Technologies, catalog #C10418). The mean fluorescence intensity of the EdU⁺

**Fig. 6 | SIRT5 activates TKT by mediating its demalonylation.**
**a**, **b** Immunofluorescent staining results for FLAG-SIRT5 WT/H158Y (in green) and TKT (in red). Yellow in the merged magnified images indicates co-localization. Scale bar, 5 µm (**a**). Fluorescence intensity of FLAG-SIRT5 WT/H158Y (green line) and TKT (red line) traced along the white line in HCT116 and LoVo cells using the line profiling function of ImageJ (**b**). This figure represents three independent experimental replicates with similar results. **c** Endogenous SIRT5 was immunoprecipitated with the anti-SIRT5 antibody, followed by Western blotting using an anti-TKT antibody in HCT116 and LoVo cells. The control comprised immunoprecipitation with IgG. **d** The interaction between FLAG-SIRT5 WT/H158Y and TKT in HCT116 and LoVo cells. **e** Malonylation (MalK) levels of exogenous TKT in HCT116 and LoVo cells expressing the control vector, SIRT5 WT, or SIRT5 H158Y. **f** The MalK levels of exogenous TKT in SIRT5-deficient HCT116 and LoVo cells were determined by Western blotting. **g**, **h** HA-tagged TKT proteins were purified using immunoprecipitation and incubated with different concentrations of malonyl-CoA (0, 1, and 2 mM) at 37 °C for 60 min. TKT activity was determined. Representative images (**g**) and quantification (**h**) of TKT activity. ($n = 3$ biologically independent experiments). **i** K281 of TKT is evolutionarily conserved across species. These sequences of TKT from humans to Gallus gallus were aligned. **j** HA-tagged TKT WT/K281R/K282R/K283R mutants were transfected into HCT116 cells, followed by treatment with SIRT5 siRNAs. TKT was immunoprecipitated and MalK levels were determined. **k** HCT116 cells expressing HA-tagged TKT WT/K281R mutant were treated with or without SIRT5 siRNAs. TKT activity was measured and normalized against protein levels. ($n = 3$ biologically independent experiments). Values in (**h** and **k**) represent the mean ± SD. $P$ values were calculated using one-way ANOVA with Tukey's multiple comparisons test. ns, not significant. Source data are provided as a Source Data file.

population was then analyzed using the FlowJo software (version 10, TreeStar).

## Soft agar colony formation assay

This assay was performed 48 h after transfection with SIRT5 siRNAs. Cells were counted and seeded in six-well plates ($10^4$) in a layer of 0.7% agar/complete growth medium over a layer of 1.2% agar/complete growth medium. A cell medium containing the indicated concentration of A, U, C, and G was then replenished every 3 days. Cultures were grown in a humidified incubator at 37 °C. Twenty-one days after seeding, cells were incubated with 4% paraformaldehyde for 10 min, and 0.5% crystal violet was used for staining cell colonies. Thereafter, megascopic colonies were counted under a light microscope. Colonies with a diameter of >50 µm were counted and analyzed.

## Measurement of enzymatic activity

Cells were homogenized with ice-cold 0.1 M Tris-HCl buffer (pH 7.6). After centrifugation, the supernatant was collected. Protein concentration was then determined using the BCA Protein Assay Kit and recorded as C (g/L). TKT activity was measured as previously described by ref. [54], with minor modifications. Briefly, the supernatant (50 µL) was mixed with a 200 µL reaction mixture containing 14.4 mmol/L R5P, 190 µM/L NADH, 380 µM/L TPP, > 250 U/L glycerol-3-phosphate dehydrogenase, and >6500 U/L triose phosphate isomerase. The optical density of TKT was then immediately measured at 340 nm, and then once every 5 min for 1 h. Moreover, TKT activity was deduced from the difference in absorbance measured at 15 and 45 min. The enzymatic activity assay was repeated three times for each group. TKT activity (%) of the treatment group was then normalized to that of the control group (100%).

## Immunofluorescence

HCT116 or LoVo cells were plated in four-well chamber slides and transfected with siRNA or plasmids, as indicated. Cells were fixed in 4% formaldehyde, permeabilized with 0.2% Triton X-100, and blocked in 1% BSA in PBS for 1 h at room temperature. They were then stained with rabbit polyclonal anti-γH2A.X (1:400), rabbit polyclonal anti-TKT (1:100), rabbit polyclonal anti-TALDO1 (1:100), rabbit anti-phospho RPA32 (S4/S8) (1:200), mouse monoclonal anti-RPI (1:100), mouse monoclonal anti-RPE (1:100), mouse monoclonal anti-FLAG (1:1200), rabbit polyclonal anti-SIRT5 (1:100, Sigma), and mouse polyclonal anti-SIRT5 (1:100, Santa Cruz) antibodies, followed by secondary antibodies (donkey anti-mouse DyLight 488 and donkey anti-rabbit DyLight 594). After staining with DAPI (1:10000), the media chamber was removed from the glass slide, treated using Prolong Gold anti-fade reagent (P7481), and sealed with a coverslip. Thereafter, fluorescence was analyzed using a confocal laser scanning microscope fitted with a 63× oil immersion objective (Carl Zeiss, AG, Germany). The Zeiss digital image processing software, ZEN® (blue edition) was used. Thin optical sections with optimal intensity were assessed. Finally, co-localization

analysis was performed using the "colocalization" module in ImageJ. For fluorescence microscopy of γ H2AX staining, the percentage of positive cells (>10 foci) out of 100 cells for each sample was calculated.

## Immunoprecipitation

Cells were seeded on 10 cm dishes and exposed to various treatments. Then the harvested cells were lysed with RIPA buffer (50 mM Tris–HCl pH 7.5, 150 mM NaCl, 0.1% sodium deoxycholate, 0.1% SDS, 1 mM EDTA pH 8.0, 1% NP-40) containing proteinase and phosphatase inhibitor cocktail (Thermo Fisher) and incubated on ice for 10 min. Cell lysates were centrifuged at 4 °C using a benchtop centrifuge (5417 C, Eppendorf) to remove debris. Genomic DNA was eliminated with DNase (79254, QIAGEN) at room temperature for 10 min. 40 µL of the lysate was transferred to a new tube, boiled in SDS sample buffer without DTT for 5 min at 95 °C, and saved as input proteins at −20 °C. 1 µg primary antibody was added to cell lysate, which was incubated at 4 °C overnight with gentle rotation. Protein G Agarose (20398, Thermo Fisher) was blocked in 1% BSA in TBS (overnight at 4 °C on a rotating platform). Protein G Agarose beads were added to the mixture of lysate and antibody at room temperature for 45 min with rotation. The beads then were washed in PBS three times and the residual liquid was removed using a syringe. The resulting beads were boiled at 95 °C for 5 min in 35 µL SDS sample buffer. Both the input and immunoprecipitated samples were subjected to western blot analysis.

## Comet assay

HCT116 and LoVo cells were transfected with SIRT5 or NC siRNA for 48 h and then cultured with a mixture of four nucleosides (10 µM) for 16 h. DNA damage was then measured by performing comet assay (single-cell gel electrophoresis) using a comet assay kit (Cell Biolab, San Diego, CA, USA), according to manufacturer instructions. Briefly, $1 \times 10^5$ cells/mL were re-suspended in ice-cold PBS (without $Mg^{2+}$ and $Ca^{2+}$). Thereafter, the samples were mixed with comet agarose at a ratio of 1:10 (v/v), homogenized by pipetting, and then this mixture (75 µL/well) was immediately transferred onto the OxiSelect™ Comet Slide. The slide was horizontally maintained and carefully transferred from the alkaline solution to a horizontal electrophoresis chamber. Electrophoresis was then performed in an alkaline buffer. The assay was performed under low/dim light conditions to avoid damage to the samples by ultraviolet light. Subsequently, the slides were then stained with a Vista Green DNA dye, and visualized by fluorescence microscopy. In addition, measurement was performed using a public domain PC-image analysis program CASP software v1.2.2 (CASPLab, University of Wroclaw, Wroclaw, Poland). This assay was repeated at least three times, and extent of DNA damage was determined by measuring the tail moment of 100 individual comets.

## Plasmids, adenovirus, and lentiviral particle construction

The control, FLAG-SIRT5 WT, and FLAG-SIRT5 H158Y-overexpression plasmids were constructed by Genechem (Shanghai, China). To avoid

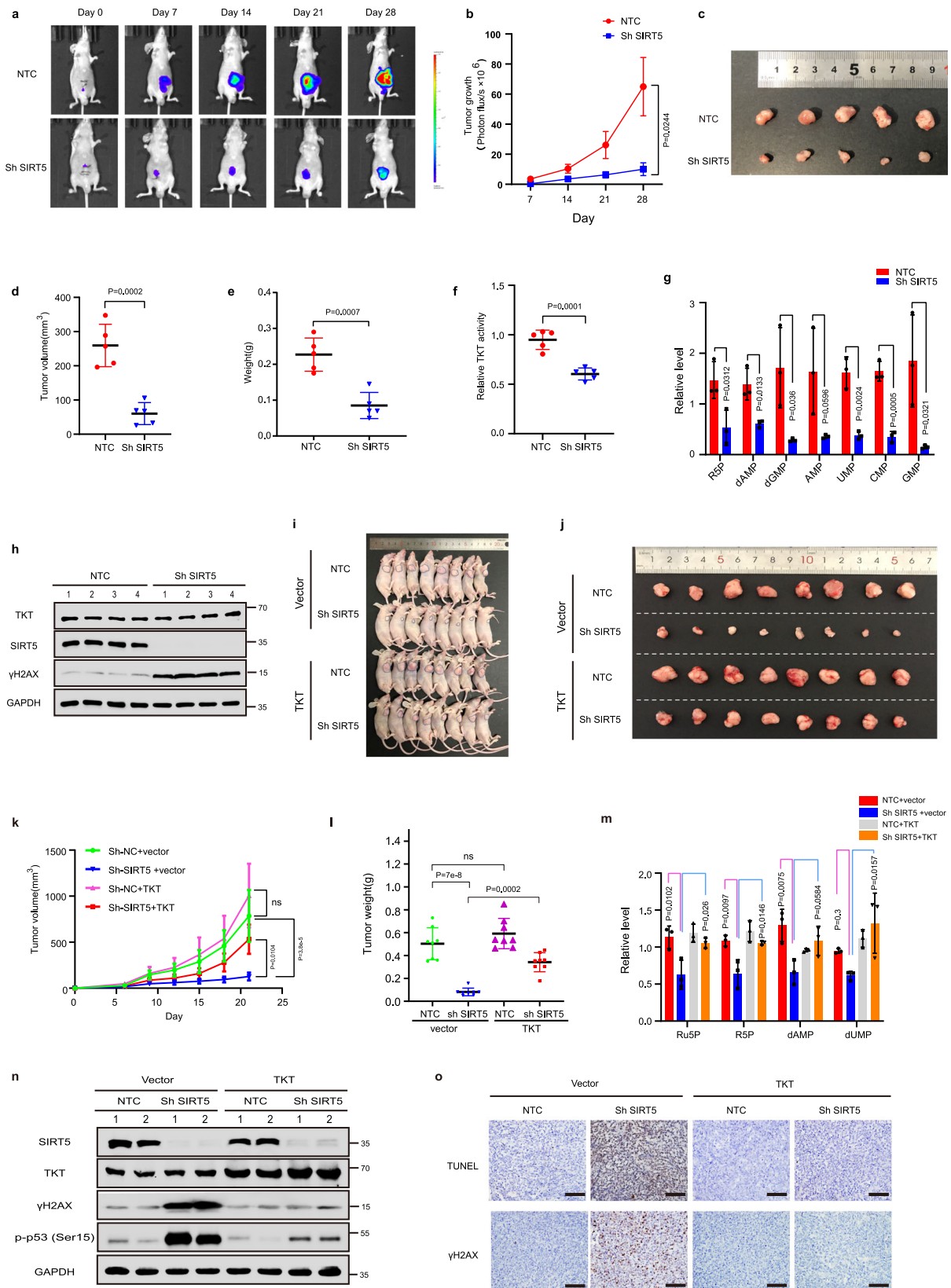

any effects on SIRT5 localization, FLAG tags were placed at the C-terminal end of SIRT5 plasmids, and this did not affect the mitochondrial localization signal. TKT WT/K281R/K282R/K283R plasmids were constructed by Generay Biotechnology (Shanghai, China). Adenovirus SIRT5 shRNAs, adenovirus NTC shRNA, lentiviral NTC shRNA, lentiviral SIRT5 shRNAs, and TKT overexpression adenovirus were

constructed by Obio Technology Company (Shanghai, China). Adenovirus serotype 5 was the source and nature of adenoviruses.

## Stable cell line generation

Stable SIRT5-expressing cells with the control vector, and FLAG-SIRT5, H158Y were established as previously described[24]. Lentiviral particles

**Fig. 7 | SIRT5 promotes CRC growth by activating TKT to sustain the nucleotide pool in vivo. a, b** A surgical orthotopic mouse model by injecting luciferase-transfected HCT116 cells stably expressing the non-target control (NTC) shRNA or SIRT5 shRNAs into the cecum of nude mice was generated. A bioluminescence imaging system was used to monitor tumor growth weekly, and mice were sacrificed 4 weeks after tumor implantation (**a**), measurements (photons/s) of tumor volume using live bioluminescence imaging at indicated times (**b**). (*n* = 5 mice per group). Values represent mean ± SEM. **c** Tumors from the two groups were dissected and photographed. (*n* = 5 mice per group). **d, e** Tumor volume (**d**) and weight (**e**) were measured on the last day of the experiment at autopsy. (*n* = 5 mice per group). **f** TKT activities in tumor lysates derived from orthotopically implanted CRC tumors were measured. (*n* = 5 mice per group). **g** R5P and nucleotide levels in orthotopically implanted tumors were measured by targeted metabolomics

analysis. (*n* = 3 mice per group). **h** Western blotting of SIRT5, TKT, and γH2AX in orthotopically implanted CRC tumors. GAPDH served as a loading control. Values indicate means ± SD in (**d–g**), compared by the two-sided Student's *t* test. **i–l** HCT116 cells under different conditions were injected subcutaneously into nude mice. Tumors from different groups were dissected and photographed (**i, j**), and tumor volume and weight were measured (**k, l**). (*n* = 8 mice per group). **m** R5P and nucleotide levels in tumor lysates derived from subcutaneous xenograft tumors. (*n* = 3 mice per group). **n** Immunoblotting of SIRT5, TKT, phospho-p53 (Ser15), and γH2AX proteins in subcutaneous xenograft tumor tissues under different treatments. **o** Representative TUNEL and γH2AX staining of subcutaneous xenograft tumors at day 21. Scale bar, 50 μm. Values represent mean ± SD. One-way ANOVA with Tukey's multiple comparisons test for (**k–m**). ns not significant. Source data are provided as a Source Data file.

carrying NTC shRNA or SIRT5 shRNAs were transfected into HCT116 and HT29 cells. After 72 h of transfection, the infected cells were selected in a medium with 1 μg/mL puromycin for 1 week. SIRT5 knockdown efficiency was determined by Western blotting, and cells showing stable SIRT5 knockdown were then further cultured in a medium supplemented with 10% fetal bovine serum and 1% penicillin-streptomycin.

## IHC

Tumors dissected from nude mouse xenograft models of CRC were subjected to IHC. Human CRC tissues were obtained from 60 patients with CRC who underwent surgery at Renji Hospital, Shanghai Jiao Tong University School of Medicine (Shanghai, China). The clinicopathological characteristics of CRC patients (such as age, gender, etc.) are summarized in Supplementary Table 1. The study was performed in accordance with the 1975 version of the Declaration of Helsinki. Ethical consent was approved by the ethics committee of Renji Hospital, Shanghai Jiao Tong University School of Medicine. All patients provided written informed consent and received no financial compensation. For immunohistochemistry, 6-μm formalin-fixed paraffin-embedded sections were incubated with antibodies against γH2AX (1:400, CST) and SIRT5 (1:400, Sigma). Validation for each primary antibody is provided on manufacturer websites. HRP-conjugated secondary antibodies were added, followed by mounting with diaminobenzidine. The pathological evaluation was conducted in a blinded manner. Protein expression levels were quantified using a visual grading system based on the extent (percentage of positive tumor cells) and intensity of staining. SIRT5 and TKT protein expression levels were assessed depending on staining intensity and extent under a microscope (200×). Staining intensity score was evaluated on a scale of 0–3: 0 = no staining, 1 = weak staining, 2 = moderate staining, and 3 = strong staining. Further, staining extent score indicated the percentage of positively stained cells (0 = 0%–5%, 1 = 6%–25%, 2 = 26%–50%, 3 = 51%–75%, and 4 = 76%–100%). To derive the final score (protein expression), staining intensity and extent scores were multiplied. For further analyses, this multiplication product (the corresponding score) was used to define the cutoff value for different protein expression levels. In the correlative study, the expression levels of relevant proteins were classified into two categories. Tissue apoptosis was analyzed using a terminal deoxynucleotidyl transferase nick-end-labeling (TUNEL) staining kit (Keygen Biotech, Nanjing, China), according to manufacturer instructions.

## Measurement of metabolite levels

For stable isotope tracing analysis, LoVo cells (2 × 10⁶/sample) were grown to 80% confluence in complete media. The cells were briefly rinsed twice with PBS and the medium was then replaced with glucose-free Dulbecco's modified Eagle medium supplemented with 11.1 mM [1,2-$^{13}$C$_2$]-glucose, 2 mM label-free glutamine in 10% dialyzed fetal bovine serum, 100 U mL$^{-1}$ penicillin–streptomycin, and 3.7 g/L sodium bicarbonate, followed by incubation for 0.5, 1, 6, 12, and 24 h.

Subsequently, cell metabolites were extracted by adding pre-cold 80% (v/v) methanol and centrifuged at 15000 *g* for 15 min at 4 °C. The supernatant was evaporated until dry, and the residue was reconstituted in 100 μL 50% aqueous acetonitrile (1:1, v/v) prior to UHPLC–HRMS/MS analysis. Chromatographic separation was performed on a Thermo Fisher Ultimate 3000 UHPLC system with a Waters BEH Amide column (2.1 mm × 100 mm, 1.7 μm). The injection volume was 2 mL, the flow rate was 0.4 mL/min, and the column temperature was 10 °C. The mobile phases consisted of water (phase A) and acetonitrile/water (90:10, v/v) (phase B), both with 15 mM ammonium acetate (pH = 9, modified using ammonium hydroxide). Linear gradient elution was performed using the following program: 0 min, 95% B and held to 2 min; 5 min, 85% B; 7 min, 80% B; 11 min, 75% B; 12 min, 55% B and held to 13.5 min; and 14 min, 95% B and held to 18 min. The eluents were then separately analyzed using Thermo Fisher Q Exactive™ Hybrid Quadrupole-Orbitrap™ MS (QE) in the heated electrospray ionization negative ion mode. For data processing of the raw data, Xcalibur 4.0 was used. For stable isotope tracing analysis, the measured distribution of mass isotopomers was corrected based on the natural abundance of isotopes using the IsoCor software (Software Version 2.2.0).

UHPLC–tandem MS (UHPLC–MS/MS) analysis was performed on an Agilent 1290 Infinity II UHPLC system interfaced with an Agilent 6470 A Triple Quadrupole MS (QQQ). To ensure that the same amount of sample is used for targeted metabolomics analysis, 1 × 10⁷ HCT116 cells/sample or 50 mg of frozen tissue were taken. The cell sample in 80% aqueous methanol was subjected to three cycles of ultrasonication for 1 min and interval for 1 min in a ice-water bath. After incubation at −20 °C for 30 min and centrifugation at 15,000 *g* for 15 min at 4 °C, the supernatant was dried and reconstituted in 100 μL of 50% aqueous acetonitrile containing 2 μg/mL ATP-$^{13}$C$_{10}$ as an internal standard, followed by UHPLC–MS/MS. The quality control sample was obtained by isometrically pooling all prepared samples. All standards were separately prepared and mixed to form a standard solution containing 20 μg/mL nucleotides. This mixed standard solution was serially diluted and finally mixed isometrically with internal standards (4 μg/mL ATP-$^{13}$C$_{10}$) to obtain a standard curve. The samples were injected onto a Waters UPLC BEH Amide column (100 mm × 2.1 mm, 1.7 μm) at a flow rate of 0.25 mL/min. The mobile phase consisted of (A) water and (B) 90% aqueous acetonitrile, both with 15 mM ammonium acetate (pH = 9). Chromatographic separation was performed using a gradient elution program: 0–1 min, 90% B; 1–4 min, 90% B–85% B; 4–8 min, 85% B–80% B; 8–15 min, 80% B–65% B; 15–15.2 min, 65% B–40% B; 15.2–16.9 min, 40% B; and then back to initial gradient at 17.1 min and equilibrated for 20 min. The eluted analytes were ionized in an electrospray ionization source in the positive mode. Raw data were processed with Agilent MassHunter Workstation (vB.08.00) using default parameters and assisting manual inspection to ensure qualitative and quantitative accuracies of each compound. A standard curve was constructed for nucleotides standard and used to determine nucleotides concentration of each unknown sample.

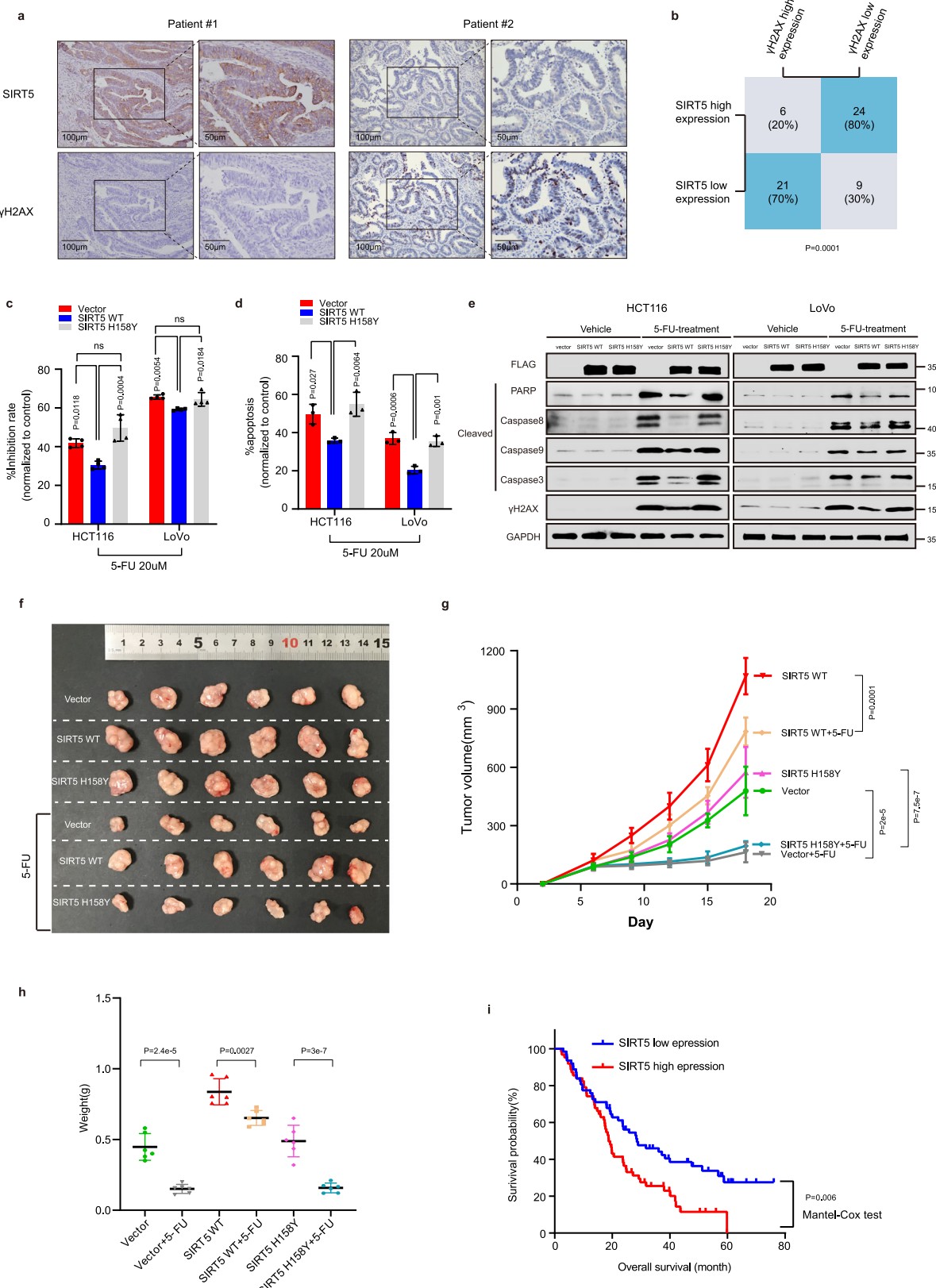

## Reactive oxygen species (ROS) assay

Intracellular ROS levels were detected in control and SIRT5 siRNA-transfected cells at 48 h, followed by exposure to 10 μM DCFDA probe (Abcam) for 30 min. Cells treated with 10 μM hydrogen peroxide (H₂O₂) for 30 min served as a positive control. ROS production was measured by flow cytometry. SIRT5 silencing-induced ROS levels were assessed by monitoring an increase in fluorescence.

## 8-hydroxy-2′-deoxyguanosine (8-OH-dG) detection

The level of 8-OH-dG, a marker of DNA/RNA oxidative damage, was measured in HCT116 and LoVo cells at 48 h post-transfection with

**Fig. 8 | Levels of SIRT5 correlates with γH2AX and predicts outcomes in patients with CRC. a** Representative immunohistochemistry images of SIRT5 (top) and γH2AX (bottom) in CRC tissues. **b** Statistical analysis of SIRT5 and γH2AX staining in 60 CRC tissues. Statistical significance was assessed using the chi-square test. **c** Cell inhibition rate in CRC cells stably expressing the control vector, SIRT5 WT and SIRT5 H158Y treated with or without 5-FU (20 μM). Cell inhibition rate was calculated as 100% × (control group values − experimental group values)/(control group values - blank values). (*n* = 4 biologically independent experiments). **d**, **e** Apoptosis was detected by flow cytometry (**d**, *n* = 3 biologically independent experiments) and immunoblotting (**e**) in HCT116 and LoVo cells stably expressing the control vector, SIRT5 WT, or SIRT5 H158Y, cultured with or without 5-FU

(20 μM). **f** Representative data of tumors in nude mice bearing HCT116 cells with different treatments. **g**, **h** Statistical analysis of tumor growth curves (**g**) and tumor weight (**h**) in different groups. (*n* = 6 mice per group). Values represent mean ± SD. Statistical significance was determined by one-way ANOVA with Tukey's multiple comparisons test (**c**, **d**, **g**, and **h**). **i** Overall survival (OS) was compared between the low-SIRT5 group (*n* = 62 human samples) and the high-SIRT5 group (*n* = 62 human samples) of CRC patients treated with FOLFOX or FOLFIRI. *p* value was calculated by the Mantel–Cox test. Gene expression data can be found online (Gene Expression Omnibus, accession no. GSE72970). ns, not significant. Source data are provided as a Source Data file.

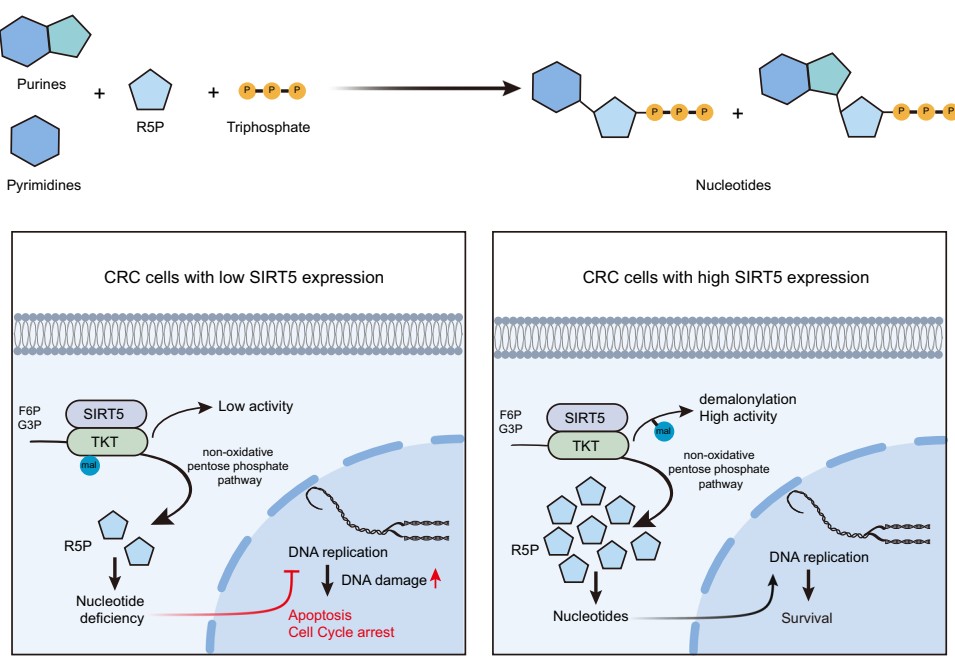

**Fig. 9 | Schematic model showing that SIRT5 promotes the non-oxidative PPP by interacting with TKT, resulting in TKT demalonylation and activation.** This maintains intracellular R5P and nucleotide levels, which supports CRC cell survival.

control siRNA or SIRT5 siRNAs using an ELISA kit from Abcam (ab201734, USA). Cells treated with 10 μM $H_2O_2$ for 30 min served as a positive control. Briefly, DNA was purified from cell lysates using a commercial extraction kit. The sample was then digested with nuclease P1 and incubated with alkaline phosphatase for 30 min at 37 °C. 8-OH-dG standard, zero standard, and digested DNA samples were added to a plate along with diluted 8-OH-dG antibody, followed by incubation at room temperature for 1 h. TMB substrate reaction was performed in the dark at room temperature for 30 min. Absorbance at 450 nm was finally measured on an ELISA plate reader.

### DNA fiber analysis
Exponentially growing HCT116 cells were labeled with the medium containing 25 μM thymidine analog iododeoxyuridine (IdU; CSN12762, CSNpharm) for 30 min. Cells were then washed with PBS before the addition of warm media containing 250 μM chlorodeoxyuridine (CldU; C6891, Sigma) for another 30 min. Labeled cells were washed with PBS twice, harvested, and lysed on slides, and DNA was stretched by tilting the slides. The DNA was fixed in the 3:1 methanol/acetic acid solution and then airdried completely. Next, fibers were denatured using 2.5 M HCl for 80 min and blocked with 5% BSA. To detect the incorporation of CldU and IdU in DNA fibers, coverslips were incubated with rat anti-BrdU (ab6326, Abcam) and mouse anti-BrdU (B2531, Sigma) primary antibodies for 2 h. The slides were stained with goat anti-rat Alexa Fluor 488 (ab150157, Abcam) and goat anti-mouse Alexa Fluor 594 (ab150116, Abcam) secondary antibodies for 1 h. Slides were mounted

with an anti-fade solution and images were acquired using a confocal microscope. Fiber lengths were analyzed using ImageJ.

### Mitochondrial isolation
Mitochondrial isolation was performed using the Mitochondrial Isolation kit (Beyotime). CRC cells were digested, collected, and incubated on ice in Mitochondrial Isolation buffer added with PMSF for 15 min. After homogenization, the homogenates were separated into mitochondrial fractions and cytoplasmic fractions by discontinuous density gradient centrifugation at $600 \times g$, $11,000 \times g$, and $12,000 \times g$ at 4 °C. The isolated cytoplasm and mitochondria were used for subsequent immunoprecipitation (IP) experiments.

### Immunoelectron microscopy
The CRC cells were cultured on 3 mm sapphire discs. An aluminum planchette with 25 μm depth inner space was used as a cover. The samples were frozen immediately using the EM ICE high-pressure freezing machine (Leica) and transferred into the EM ASF2 (Leica) for substitution. Next, the samples were incubated for 48 h in acetone containing 0.2%UA at −90 °C. Then the temperature was raised to −50 °C in 4 h. After being incubated in acetone containing 0.2%UA for another 12 h, the temperature was raised to −30 °C in 4 h. After incubation for 2 h at −30 °C, the samples were rinsed three times with pure acetone (15 min each). Then the samples were gradually infiltrated in HM20 resin with grades of 25%, 50%, 75%, and pure resin (2 h each) at −30 °C. After being infiltrated in pure resin overnight, the samples

were embedded in gelatin capsules. The samples were polymerized under UV light for 48 h at −30 °C and 12 h at 25 °C. After polymerization, the samples were trimmed and ultra-thin sectioned with a microtome (Leica UC7). Serial thin sections (100 nm thick) were collected onto formvar-coated nickel grids. Immunostainings were processed as follows: The grids with sections were incubated in 0.01 M PBS containing 1% BSA, 0.05% Triton X-100 and 0.05% Tween20 for 5 min. Then the sections were incubated in the mouse anti-TKT (ab112997, Abcam) diluted in 0.01 M PBS containing 1% BSA and 0.05% Tween20 at 4 °C overnight. After being washed six times (2 min each) with PBS, the sections were incubated in the secondary antibody goat anti-mouse conjugated with 10 nm gold (ab39619, Abcam) diluted in 0.01 M PBS containing 1% BSA and 0.05% Tween20 (1:50) for 2 h at RT. PBS for 2 min (six times) and distilled water for 2 min (four times). After having been dried in the air, sections were examined in transmission electron microscopy (Thermo Fisher/FEI Talos L 120 C). Stained samples without the primary antibody were used as negative controls.

### Statistics and reproducibility

Unpaired two-tailed Student's $t$ test was used for comparison of two groups with normal distribution. One-way ANOVA corrected with Tukey's multiple comparisons test was used for the comparison of three or more experimental conditions. The correlation between the expression of SIRT5 and TKT was analyzed using the $\chi 2$-test. $p < 0.05$ indicated statistical significance. If not otherwise mentioned, each experiment was repeated independently with similar results at least three times.

### Reporting summary

Further information on research design is available in the Nature Research Reporting Summary linked to this article.

## Data availability

The metabolites data for Fig. 2a are available in figshare with the identifier (data https://doi.org/10.6084/m9.figshare.5731485). The clinical data about the relevance of SIRT5 expression with chemoresistance for Fig. 8i were retrieved from GEO database with the code GSE72970. Source data are provided in this paper. Source data are provided with this paper.

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

## Acknowledgements

This study was supported by the National Natural Science Foundation of China (No. 81972203, 81772506, 81530072, 81830081, 81902362), the funds from Shanghai Shenkang Center (SHDC12018121), Shanghai Jiao tong University "STAR" plan (20190102), Shanghai Municipal Education Commission-Gaofeng Clinical Medicine Grant Support (No. 20152210), and Shanghai Sailing Program (No.19YF1428900). We would like to thank Prof. Ming Lei of the State Key Laboratory of Oncogenes and Related Genes (Ninth People's Hospital, Shanghai Jiao Tong University School of Medicine, Shanghai, China) for excellent technical assistance.

## Author contributions

H.-L.W. and Y.-X.C. designed the experiments, analyzed the data, and wrote the paper. H.-L.W., Y.C., Y.-Q.W., E.-W.T., J.T., Q.-Q.L, and C.-M.L performed the experiments. X.-M.T., J.H., and Q.-Y.G. supplied critical materials. J.-Y.F. reviewed the paper, and Y.-X.C. supervised the project. All authors have read and approved the final version of the paper.

## Competing interests

The authors declare no competing interests.
