## [Peer Review File · Nature Communications]

REVIEWER COMMENTS

Reviewer #1 (Remarks to the Author):

The authors performed in vitro and in vivo studies along the SIRT5-TKT-non-oxidative PPP axis in the context of CRC. They demonstrated ablation of SIRT5 led to decreased proliferation and increased apoptosis due to reduced supply of R5P and downstream nucleotides. In addition to this metabolic investigation, the authors also resolved the potential mechanism of how SIRT5 regulates TKT activity through demalonylation at a specific enzymatic site. Finally, the authors presented promising results of drug targeting SIRT5 as a cancer treatment to CRC.

The paper is overall well organized and experiments well designed to elucidate the aforementioned scientific discoveries. My comments are summarized below to further improve the manuscript.

1. It is not clear whether metabolite pools have been normalized by cell count/protein content in Figures 2a, 2b and 7g. The authors should clearly state method of normalization and use of internal metabolite standards for pool size analysis.

2. The effect of nucleosides supplementation should be shown for the experiments performed in Figures 2f-i.

3. The authors claimed that 'K281R mutation resulted to a significant reduction in malonylation as shown in Fig 6j.' However, it appears lane 1 and 3 showed little difference in malonylation.

4. The authors may need to seek grammatical advice from experts. Examples include, but not limited to, in Lines 324-325, 'The results showed that silencing SIRT resulted IN a significantLY in decreased tumor volume and weight'. The scientific part of the manuscript is pretty solid though.

5. The paper should discuss findings from these papers about TKT and non-oxidative PPP, doi.org/10.1002/aic.16423 and 10.18632/oncotarget.10429.

Minor issues: some texts are blocked by images in Figure 2c, 5f, etc. Symbols/texts are not shown correctly in Lines 592, 598, etc. There is a space issue in Line 302 (SIRT5. Lysine 281). In Line 330, it should be figure 7e, not 7E.

Reviewer #2 (Remarks to the Author):

The finding that TKT activity is regulated via lysine demalonylation is exciting and novel. However, the study would benefit from additional studies to clarify the mechanism of regulation and its relevance in CRC. The experiments in the current study do not clearly establish how SIRT5, a mitochondrial enzyme in mammalian cells, regulates a cytosolic metabolic pathway to significantly impact cancer cell growth and genome stability. The study would also benefit from in vivo studies in mouse models that are more relevant to CRC, as well as incorporation of bioinformatic data from CRC patients. Specific comments are described below:

- The authors report that SIRT5 regulates TKT, which is an enzyme located in the cytosol; while there are isoforms of SIRT5 that are reported to localize to the cytosol, these isoforms also are reported to have decreased deacylase activity compared to mitochondrially localized isoform of SIRT5, and the mitochondrially localized isoform is most highly expressed in human cell lines. In light of this, the authors should add data to Figure 1 to clarify 1) if their study focuses on a specific isoform of SIRT5 (from Figure 1a, it is unclear if the siRNA approach targets all SIRT5 isoforms?) and 2) clarify how much SIRT5 is localized in the cytosol vs mitochondria of their experiments.

- In light of the above point, the impact of the study would be improved if the authors could connect the localization pattern of SIRT5 shown in Figures 4f and 6a to the localization pattern of SIRT5 in human CRCs. For example, while SIRT5 expression is increased in CRCs, if the majority of this SIRT5 appears to be mitochondrial, this would suggest that SIRT5 regulation of TKT/the PPP may not be the predominant pathway regulated by SIRT5 in patient tumors.

- The authors should clarify whether FLAG tags on all SIRT5 plasmids (WT and mutant versions) utilized for experiments in Figures 4j-l and 6a-e were introduced at the N- or C-terminal end of SIRT5, as this is unclear in the current manuscript. If the tag is at the N-terminal end, these experiments must be repeated with a C-terminally tagged construct to ensure the tag is not affecting SIRT5 localization by masking the N-terminal mitochondrial localization signal.

- While the authors show that silencing of SIRT5 induces DNA damage in Figure 1 and diminishes nucleotide pools in Figures 2a-b, since this is an siRNA-mediated approach rather than a stable knockdown or knockout of SIRT5, it is unclear whether there would be any metabolic adaptation in these cell lines to attempt to restore nucleotide pools and protect cells from DNA damage and apoptosis. Given that cells are known to adapt flux through the oxidative versus non-oxidative PPP to respond to stress and meet metabolic demands, one might expect that over time, CRC cells and tumors with SIRT5 loss would adapt their metabolism to protect their nucleotide pools. The authors should consider complementing the experiments shown in Figures 2a-b, 2f-I, and 4h-i with studies in CRC cell lines with stable knockdown or knockout of SIRT5 that have been passaged to better understand whether any metabolic adaptation to SIRT5 loss/reduced TKT activity occurs.
- The authors could increase the impact and clinical relevance of their findings by including treatment in their cell lines and xenograft model with a DNA-damaging chemotherapy challenge. A chemotherapy challenge could easily be incorporated in the experiments shown in Figures 4I and 7b-g. Given the finding that SIRT5 preserves nucleotide pools and protects cells from DNA damage, it's reasonable to hypothesize that cells lacking SIRT5 should be more sensitive to DNA-damaging chemotherapy (and perhaps overexpression of SIRT5 is more associated with chemoresistance – is there bioinformatic data available to support this?).
- In Figure 7, the authors do not demonstrate that they can rescue the xenograft phenotype either with SIRT5 expression or nucleotides, and the authors would further strengthen the findings of Figure 7 by including SIRT5 overexpression as well, given that SIRT5 is often overexpressed in CRCs, not lost. The authors should also include the SIRT5 catalytic inactive mutant CRC cell lines in their xenograft studies to more strongly connect SIRT5 activity and regulation of TKT activity in vivo.
- Both cell lines the authors use in this study have oncogenic KRas mutations; is KRas mutation required to observe the metabolic effects the authors see in the study, or is this regulation of the PPP/TKT in cells intrinsic to SIRT5 regardless of the presence of additional driver mutations? Inclusion of experiments throughout the study in a CRC cell line with WT Kras (HT29, SW48) or another cell line would shed light on the relevance/contribution of oncogenic driver mutations to the observed SIRT5-associated phenotypes. This is especially relevant to consider since studies in SIRT5 KO mice in the absence of additional challenge/mutations have not revealed any strong phenotypes.
- To more strongly establish the SIRT5-TKT axis as an attractive therapeutic target in CRC, the study would benefit from in vivo studies in a more cancer-relevant mouse model than the xenograft studies shown in Figure 7. Either a GEMM with oncogenic driver mutations present (Apc, Kras) where SIRT5 could be knocked out or orthotopic injection of CRC tumor cells, would be a better model for the in vivo experiments.
- The conclusions of the study regarding the relevance of SIRT5-mediated regulation of TKT activity in CRC would be strengthened by analyzing TKT activity from either GEMM tumor samples where SIRT5 was knocked out, or CRC patient tumor samples.

Reviewer #3 (Remarks to the Author):

In the present manuscript, Wang and colleagues studied the role of sirtuin 5 (SIRT5) in CRC metabolism and proliferation. SIRT5 is a mitochondrial enzyme that catalyzes the removal of PTMs including acetylation, succinylation and malonylation in a NAD⁺-dependent manner. Here, different CRC models, including two CRC cell lines, a CRC xenograft mouse model as well as human colorectal tumor specimens were used to detail the role of SIRT5. Using siRNA-mediated knockdown of SIRT5, the authors showed reduced levels of ribose-5-phosphate (5RP) and, consequently, lower levels of dNTPs. This resulted in the formation of DNA damage and cell death induction in CRC cell lines, although the exact mechanism remains to be shown. Interestingly, the authors revealed a link between transketolase (TKT), an enzyme required in the non-oxidative

pentose phosphate pathway, and SIRT5. They provided mechanistic evidence that SIRT5-mediated demalonylation of TKT promotes TKT activity and thus R5P production, which is necessary for the proliferation of CRC cells. These findings were translated into an in vivo CRC xenograft model, corroborating the in vitro results.

In general, this is an interesting study, which provides novel insights into the metabolic deregulation of CRC cells as potential target for CRC therapy. Nevertheless, I see some major issues, which are particularly important for the translational aspects and the scope of this study.

1.) The authors showed previously (Wang et al. 2018 Nat Commun) that SIRT5 is differentially expressed on the protein level among CRC cell lines and non-cancerous cells (FHC, fetal human colonocytes) with a factor of 2-4.

It is therefore crucial to address the role of SIRT5 (and TKT) also in non-cancerous colonocytes, such as FHC or HCEC, in order to elaborate whether the described mechanism/SIRT5 phenotype is a hallmark in CRC cells or also observed in normal colonocytes.

2.) Another limitation of this study is the use of only two CRC cells lines (HCT116 and LoVo cells, both p53 WT, MSI). Thus, key experiments have to be repeated in other CRC cells lines with MSS and p53 mutations (e.g. HT29 cells, SW480 cells). This will clearly extend the scope of this study.

3.) The precise mechanism of DNA damage induction, particularly the role of replication stress, have to be detailed. An important question is also whether SIRT5 knockdown causes DNA damage (and cell death) in normal colonocytes. Including a positive control such as the anticancer drug hydroxyurea would further strengthen the study, since HU also results in dNTP depletion.

4.) The material & methods part lacks important information and complete sections referring to experiments shown in the main figures and SI figures. This has to be revised.

Specific comments:

Fig. 1: It is important to include p53-mutated CRC cells and normal, non-malignant colonocytes (see above)

Fig. 1B: please provide images with higher magnification to assess nuclear gH2AX staining. Please note that both in the right panel (HCT116) as well as in the left panel (LoVo) apoptotic cells are clearly visible (condensed nuclei in DAPI channel). It is known that gammaH2AX could also arise due to apoptosis. Therefore, it is recommended to check the contribution of apoptosis-mediated gH2AX formation to the overall gH2AX levels by using pan-caspase inhibitors.

Fig. 1D and E: The authors showed the formation of DNA strand breaks using the alkaline Comet assay. However, what is the underlying mechanism? It is strongly recommended to analyze further markers of replication stress (e.g. RPA foci or DNA fiber assay). Including HU as positive control would also make sense (see comment above).

Fig. 1F: It is not mentioned after which time point the DDR markers were analyzed. Furthermore, a time course experiments similar to fig. 1a would help to elucidate the time kinetics (i.e. transient or persistent DDR activation) and provide some clues on the mechanism.

Fig. 2C and D: an alkaline Comet assay would strengthen the notion that supplementation with nucleosides can reverse DNA damage

Fig.3: please include normal colonocytes and p53-mutated CRC cell lines to allow for a more general conclusion and to address the point of tumor selective cell death. DNA damage typically engages p53 as a major trigger of apoptotic cell death, thus p53-mutated cells have to be considered.

Fig. 1-3: A pharmacological approach using SIRT5 inhibitor 1 could nicely cross-validate key findings obtained by SIRT5 knockdown.

Fig. 4f: Based on the microscopy images, both TKT and SIRT5 show cellular pan-staining and no organelle-specific staining (i.e. SIRT5 in mitochondria). Could the authors please comment on that? Furthermore, a technical question came up: are these thin optical sections or z-stacks with maximum intensity projection shown here? This might critically impact the conclusions.

Fig. 4m: the authors mentioned that the observed lower gH2AX levels in HCT116 cells with WT SIRT5 overexpression after CPT treatment could be a result of increased DSB repair due to higher dNTP pools. This is very speculative and should be removed (or addressed experimentally). The expectation would rather be the opposite: given that SIRT5 overexpression causes higher dNTP levels and thus promotes DNA replication, the effects of CPT should be stronger. The TOPO I poison CPT is active in replicating cells, causing a collision of the replication fork with the cleavage complex formed by DNA, TOPO I and CPT.

Fig. 7e: Another genotoxicity marker, such as p53, should be detected to corroborate the notion that SIRT5 k.o. causes increased levels of DNA damage in tumor tissue.

Fig. 7f: please perform quantitative evaluation of gH2AX levels either by WB or IHC

Fig. 7h: Can the authors please detail on the antibody validation and the evaluation of the stainings? Further, how was distinguished between staining of tumor cells and tumor stroma?

M&M part:

- Cell culture: how was mycoplasma contamination tested?
- Western blots: how was chemoluminescence measured (imager?) and was densitometric evaluation of blots performed?
- Confocal microscopy: z-stacks or single optical sections? Which cLSM was used?
- Comet-Assay: DNA staining used? How many comets were assessed per slide/sample?
- IHC: how were the tissues collected and fixed? FFPE tissue and microtome sectioning or snap-frozen tissue and cryostat sectioning?
- In vivo models: source and nature of adenoviruses targeting SIRT5 and TKT?
- Source of human normal and tumor tissue? Ethics statement? Animal experiments approval number?
- Plasmids and constructs used?
- ROS assay and 8-OH-dG detection performed in Fig. S1: no information provided!

Point by point response to reviewers' comments

Reviewer #1 (Remarks to the Author):

The authors performed in vitro and in vivo studies along the SIRT5-TKT-non-oxidative PPP axis in the context of CRC. They demonstrated ablation of SIRT5 led to decreased proliferation and increased apoptosis due to reduced supply of R5P and downstream nucleotides. In addition to this metabolic investigation, the authors also resolved the potential mechanism of how SIRT5 regulates TKT activity through demalonylation at a specific enzymatic site. Finally, the authors presented promising results of drug targeting SIRT5 as a cancer treatment to CRC. The paper is overall well organized and experiments well designed to elucidate the aforementioned scientific discoveries. My comments are summarized below to further improve the manuscript.

1. It is not clear whether metabolite pools have been normalized by cell count/protein content in Figures 2a, 2b and 7g. The authors should clearly state method of normalization and use of internal metabolite standards for pool size analysis.

Response: We apologize for the lack of clarity and thank you for your valuable suggestion. Given that cell count or sample amount is commonly considered as an appropriate normalizer¹, we took 1×10^7 HCT116 cells or 50 mg frozen tissue to ensure that the same amount of sample was subjected to metabolomics analysis. For targeted metabolomics profiling, isotopically labeled internal standards were used to maintain system stability². With regard to revised Figure 2b and 7g, nucleotides were quantified using calibration curves prepared with dilution series of ultrapure standards (Sigma) and a labeled stable isotope ($2 \mu\text{g/mL ATP-}^{13}\text{C}_{10}$) as an internal standard. For untargeted metabolomics analysis (revised Figure 2a), metabolite levels in the control and SIRT5 siRNA-transfected cells were quantified by normalization to constant sum, with 3-chloro-L-phenylalanine as an internal standard. The Methods section has been extensively modified to reflect these changes.

2. The effect of nucleosides supplementation should be shown for the experiments

performed in Figures 2f-i.

Response: Thank you for your valuable suggestions. We assessed the effects of nucleoside supplementation on SIRT5-induced DNA synthesis. HCT116 and LoVo cells were transfected with SIRT5 siRNAs for 48 h and cultured with a mixture of the four nucleosides (10 μ m) for 16 h. Immunofluorescence and flow cytometry revealed that SIRT5 knockdown-induced inhibition of DNA synthesis was reversed by exogenous nucleoside supplementation (revised Figure 2i-l).

3. The authors claimed that 'K281R mutation resulted to a significant reduction in malonylation as shown in Fig 6j.' However, it appears lane 1 and 3 showed little difference in malonylation.

Response: To identify SIRT5-targeted acylation site(s) of TKT, we generated three TKT mutants, i.e., K281R, K282R, and K283R, in which K281, K282, and K283 were replaced with arginine, respectively. We further repeated the experiment at least three times. Immunoprecipitation analysis demonstrated that unlike cells with wild-type (WT) TKT and K282R/K283R mutation, those with K281R mutation displayed a significant reduction in malonylation level and a negligible response to SIRT5 knockdown (revised Figure 6j and Supplementary Figure 1 for reviewers).

Supplementary Figure 1 for reviewers. HA-tagged TKT WT and

K281R/K282R/K283R mutants were transfected into HCT116 cells, followed by SIRT5 siRNAs treatment. TKT was immunoprecipitated and malonylation (MalK) levels of TKT were determined by Western blotting. Integrated density values were calculated using ImageJ. Values represent mean \pm SD of three independent experiments. *p* values were calculated by ANOVA with Tukey's test. **p* < 0.05, ***p* < 0.01, ns = not significant for the indicated comparison.

4. The authors may need to seek grammatical advice from experts. Examples include, but not limited to, in Lines 324-325, 'The results showed that silencing SIRT5 resulted in a significantly in decreased tumor volume and weight'. The scientific part of the manuscript is pretty solid though.

Response: We sincerely apologize for such language errors. To improve language quality and readability, we got the manuscript edited by a native English speaker. In the revised manuscript, corrections are marked in red.

5. The paper should discuss findings from these papers about TKT and non-oxidative PPP, doi.org/10.1002/aic.16423 and 10.18632/oncotarget.10429.

Response: We appreciate you sharing these resources on TKT and non-oxidative PPP. We have referred to these manuscripts and added some additional information in the Discussion section:

"Our findings suggested that SIRT5 silencing specifically inhibited the non-oxidative PPP, indicating that SIRT5 might be a promising therapeutic target. TKT is the key enzyme in the non-oxidative PPP and controls the ratio of the oxidative PPP versus non-oxidative PPP flux. Cancer cells evidently increase the metabolic flux of the non-oxidative PPP for R5P synthesis via TKT under hypoxia³. Another study reported that active TKT maintains the non-oxidative PPP flux to generate R5P for nucleic acid synthesis⁴. Herein we observed that SIRT5 increased the non-oxidative PPP flux by activating TKT in a PTM. However, this might not be the only mechanism to upregulate or activate TKT. TKT expression is also reportedly upregulated at the transcriptional level to promote the non-oxidative PPP⁵.

Minor issues: some texts are blocked by images in Figure 2c, 5f, etc. Symbols/texts are not shown correctly in Lines 592, 598, etc. There is a space issue in Line 302 (SIRT5. Lysine 281). In Line 330, it should be figure 7e, not 7E.

Response: To address your concerns, we got the manuscript edited and formatted by a native English speaker. This has substantially improved the quality of language used in the revised manuscript.

REFERENCES

1. Silva LP, Lorenzi PL, Purwaha P, Yong V, Hawke DH, Weinstein JN. Measurement of DNA concentration as a normalization strategy for metabolomic data from adherent cell lines. *Analytical chemistry* 2013, **85**(20): 9536-9542.
2. Broadhurst D, Goodacre R, Reinke SN, Kuligowski J, Wilson ID, Lewis MR, *et al.* Guidelines and considerations for the use of system suitability and quality control samples in mass spectrometry assays applied in untargeted clinical metabolomic studies. *Metabolomics : Official journal of the Metabolomic Society* 2018, **14**(6): 72.
3. Ahn WS, Dong W, Zhang Z, Cantor JR, Sabatini DM, Iliopoulos O, *et al.* Glyceraldehyde 3-phosphate dehydrogenase modulates nonoxidative pentose phosphate pathway to provide anabolic precursors in hypoxic tumor cells. *AICHE Journal* 2018, **64**(12): 4289-4296.
4. Diaz-Moralli S, Aguilar E, Marin S, Coy JF, Dewerchin M, Antoniewicz MR, *et al.* A key role for transketolase-like 1 in tumor metabolic reprogramming. *Oncotarget* 2016, **7**(32): 51875-51897.
5. Mitsuishi Y, Taguchi K, Kawatani Y, Shibata T, Nukiwa T, Aburatani H, *et al.*

Nrf2 redirects glucose and glutamine into anabolic pathways in metabolic reprogramming. *Cancer cell* 2012, **22**(1): 66-79

Reviewer #2 (Remarks to the Author):

The finding that TKT activity is regulated via lysine demalonylation is exciting and novel. However, the study would benefit from additional studies to clarify the mechanism of regulation and its relevance in CRC. The experiments in the current study do not clearly establish how SIRT5, a mitochondrial enzyme in mammalian cells, regulates a cytosolic metabolic pathway to significantly impact cancer cell growth and genome stability. The study would also benefit from in vivo studies in mouse models that are more relevant to CRC, as well as incorporation of bioinformatic data from CRC patients. Specific comments are described below:

- The authors report that SIRT5 regulates TKT, which is an enzyme located in the cytosol; while there are isoforms of SIRT5 that are reported to localize to the cytosol, these isoforms also are reported to have decreased deacylase activity compared to mitochondrially localized isoform of SIRT5, and the mitochondrially localized isoform is most highly expressed in human cell lines. In light of this, the authors should add data to Figure 1 to clarify 1) if their study focuses on a specific isoform of SIRT5 (from Figure 1a, it is unclear if the siRNA approach targets all SIRT5 isoforms?) and 2) clarify how much SIRT5 is localized in the cytosol vs mitochondria of their experiments.

Response: Thank you for your detailed feedback. To clarify, in this study, we did not focus on a specific isoform of SIRT5. SIRT5 siRNAs used herein were targeted at the conserved sequences of SIRT5 isoforms 1 to 7.

It has been widely reported that SIRT5 mainly localizes in the mitochondrial matrix with a small portion existing extra mitochondrially^{6, 7, 8}. Intriguingly, in this study, immunofluorescent staining (revised Fig. 4f, g) and co-immunoprecipitation assay (Fig. 6c, d) revealed a strong co-localization between SIRT5 and TKT in HCT116 and LoVo cells. Although the PPP mainly occurs in the cytoplasm, it has been reported that TKT, a key enzyme in the PPP, is not only a cytosolic enzyme but also present in peroxisomes and at membranes of granular endoplasmic reticulum in liver parenchymal cells⁹. Moreover, TKT has been reported to show a strong nuclear localization in HCC cells¹⁰.

To detect the subcellular localization of TKT in CRC cells, mitochondrial and cytoplasmic proteins were extracted and Western blotting was then performed. As anticipated, TKT was detected in both mitochondrial and cytoplasmic fractions (Supplementary Figure 2a for reviewers). Immunofluorescent staining (Supplementary Figure 2b–d for reviewers) further confirmed the co-localization of TKT protein with Mito-track in CRC cells and tissues (an anti-mitochondria antibody). These data suggested that TKT interacts with SIRT5 in the mitochondria.

Supplementary Figure 2 for reviewers

Supplementary Figure 2 for reviewers. (a) Endogenous TKT and SIRT5 were detected in mitochondrial (mito) and cytoplasmic (cyto) fractions of HCT116 cells by immunoblot analysis. (b–d) CRC cells (b) and tissues (d) were immunostained for TKT (red) and Mito-Track (green); yellow in the merged images indicates the co-localization between TKT and mitochondria. Scale bar in b = 5 μ m and in d = 50 μ m. Fluorescence intensity of TKT (red line) and Mito-Track (green line) traced along the white line in CRC cells (b) using the line profiling function of ImageJ (c).

- In light of the above point, the impact of the study would be improved if the authors could connect the localization pattern of SIRT5 shown in Figures 4f and 6a to the

localization pattern of SIRT5 **in human CRCs**. For example, while SIRT5 expression is increased in CRCs, if the majority of this SIRT5 appears to be mitochondrial, this would suggest that SIRT5 regulation of TKT/the PPP may not be the predominant pathway regulated by SIRT5 in patient tumors.

Response: Thank you for this useful suggestion. To determine the localization of SIRT5 and TKT *in vivo*, CRC tissues were assessed using immunofluorescence and confocal microscopy. As shown in revised Figure 4f and Supplementary Figure 5g, the co-localization of SIRT5 and TKT was observed in not only CRC cells but also CRC tissues, implying that TKT interacts with SIRT5 in the mitochondria.

- The authors should clarify whether FLAG tags on all SIRT5 plasmids (WT and mutant versions) utilized for experiments in Figures 4j-l and 6a-e were introduced at the N- or C-terminal end of SIRT5, as this is unclear in the current manuscript. If the tag is at the N-terminal end, these experiments must be repeated with a C-terminally tagged construct to ensure the tag is not affecting SIRT5 localization by masking the N-terminal mitochondrial localization signal.

Response: Thank you for the valuable suggestion. The control, FLAG-SIRT5 WT, and FLAG-SIRT5 H158Y-overexpression plasmids were constructed by Genechem Technologies (Shanghai, China). The tags used in this study were all placed at the C-terminal end of SIRT5 plasmids, and this did not affect the mitochondrial localization signal (Supplementary Figure 3 for reviewers, see below). This information has been included in the revised manuscript.

Supplementary Figure 3 for reviewers

Supplementary Figure 3 for reviewers. Sequence of SIRT5 (WT and mutant versions) plasmid elements: CMV-3FLAG-MCS-EGFP-SV40-Neomycin.

• While the authors show that silencing of SIRT5 induces DNA damage in Figure 1 and diminishes nucleotide pools in Figures 2a-b, since this is an siRNA-mediated approach rather than a stable knockdown or knockout of SIRT5, it is unclear whether there would be any metabolic adaptation in these cell lines to attempt to restore nucleotide pools and protect cells from DNA damage and apoptosis. Given that cells are known to adapt flux through the oxidative versus non-oxidative PPP to respond to stress and meet metabolic demands, one might expect that over time, CRC cells and tumors with SIRT5 loss would adapt their metabolism to protect their nucleotide pools. The authors should consider complementing the experiments shown in Figures **2a-b**, **2f-l**, and **4h-i** with studies in CRC cell lines with stable knockdown or knockout of SIRT5 that have been passaged to better understand whether any metabolic adaptation to SIRT5 loss/reduced TKT activity occurs.

Response: As suggested, HCT116 and HT29 cells with stable knockdown of SIRT5 were generated. Lentiviral particles carrying non-target control (NTC) short hairpin RNA (shRNA) or SIRT5 shRNAs were transfected into HCT116 and HT29 cells. After 72 h of transfection, the infected cells were selected in a medium with 1 μ g/mL puromycin for 1 week. SIRT5 knockdown efficiency was then determined by Western blotting, and cells with stable knockdown of SIRT5 were further cultured in a medium

supplemented with 10% fetal bovine serum and 1% penicillin–streptomycin. SIRT5 shRNAs were able to efficiently knockdown the expression of SIRT5 (revised Supplementary Figure 2c and Supplementary Figure 4a for reviewers). Consistent with previous results on SIRT5 siRNAs, the stable knockdown of SIRT5 significantly decreased R5P, Ru5P, and nucleotide levels (revised Supplementary Figure 2d). Furthermore, targeted metabolomics analysis confirmed nucleotide pool deficiency when SIRT5 was stably downregulated (revised Supplementary Figure 2e). In addition, we performed 5-ethynyl-20-deoxyuridine (EdU) assay using SIRT5-deficient CRC cells and found that SIRT5 shRNAs treatment still inhibited DNA biosynthesis, which was reversed by exogenous nucleoside supplementation (revised Supplementary Figure 3e–h and Supplementary Figure 4b–e for reviewers). Moreover, the stable knockdown of SIRT5 in HCT116 and HT29 cells was found to markedly inhibit TKT activity (revised Supplementary Figure 5h and Supplementary Figure 4f for reviewers). Altogether, consistent with the effect of siRNAs-mediated SIRT5 knockdown, stable knockdown of SIRT5 significantly decreased nucleotide pool and inhibited DNA synthesis as well as TKT activity.

Supplementary Figure 4 for reviewers

Supplementary Figure 4 for reviewers. (a) Western blotting to assess SIRT5 expression in HT29 cells transduced with lentiviral particles carrying SIRT5 shRNAs or NTC

shRNA. (b–e) EdU assay involving immunofluorescent staining (b) and flow cytometry (d) of HT29 cells with stable SIRT5 knockdown after exogenous nucleoside supplementation. Scale bar, 50 μ m. Data in b and d were quantified and analyzed in (c) and (e), respectively. (f) TKT activity was assessed in HT29 cells with stable SIRT5 knockdown. The TKT inhibitor oxythiamine (OT; 20 μ M) served as a positive control. Values in c, e, and f represent mean \pm SD of three independent samples. Statistical significance was calculated using ANOVA with Tukey's test. ** $p < 0.01$, *** $p < 0.001$, **** $p < 0.0001$. ns = not significant for the indicated comparison.

- The authors could increase the impact and clinical relevance of their findings by including treatment in their cell lines and xenograft model with a DNA-damaging chemotherapy challenge. A chemotherapy challenge could easily be incorporated in the experiments shown in **Figures 4I and 7b-g**. Given the finding that SIRT5 preserves nucleotide pools and protects cells from DNA damage, it's reasonable to hypothesize that cells lacking SIRT5 should be more sensitive to DNA-damaging chemotherapy (and perhaps overexpression of SIRT5 is more associated with chemoresistance – is there bioinformatic data available to support this?).

Response: To address your feedback, HCT116 and LoVo cells stably expressing the control vector, SIRT5 WT, and SIRT5 H158Y were treated with/without 5-fluorouracil (5-FU). We observed that SIRT5 WT-overexpressing cells showed a significantly higher survival rate after 5-FU treatment in comparison to cells expressing the control vector or SIRT5 H158Y (revised Figure 8c). Further, in the SIRT5 WT group, 5-FU-induced apoptosis of HCT116 and LoVo cells was reduced (revised Figure 8d, e). We further generated a xenograft model and tested whether SIRT5 WT overexpression is closely associated with chemoresistance. Further, HCT116 cells (2×10^6 cells) stably expressing the control vector, SIRT5 WT, and SIRT5 H158Y were subcutaneously injected. Interestingly, 5-FU treatment significantly decreased tumor growth and weight in the control vector and SIRT5 H158Y groups, but this effect was not notable in the SIRT5 WT group (revised Fig. 8f–h). Collectively, these results indicated that SIRT5 WT is associated with chemoresistance to 5-FU. To decipher the clinical relevance of

SIRT5 expression with chemoresistance, we evaluated the relationship between SIRT5 expression and overall survival in the Gene Expression Omnibus dataset (raw data accessible via GSE72970) in which CRC patients were treated with FOLFOX or FOLFIRI. We found that the high levels of SIRT5 in CRC tissues were associated with shorter survival (revised Figure 8i), indicating that SIRT5 can predict poor prognosis in patients with CRC receiving chemotherapy; consequently, it can serve as a potential anticancer target.

- In Figure 7, the authors do not demonstrate that they can rescue the xenograft phenotype either with SIRT5 expression or nucleotides, and the authors would further strengthen the findings of Figure 7 by including SIRT5 overexpression as well, given that SIRT5 is often overexpressed in CRCs, not lost. The authors should also include the SIRT5 catalytic inactive mutant CRC cell lines in their xenograft studies to more strongly connect SIRT5 activity and regulation of TKT activity *in vivo*.

Response: Thank you for your insightful suggestions. To address the role of SIRT5 in regulating TKT activity *in vivo*, we established a xenograft tumor model in nude mice by subcutaneously injecting them with HCT116 cells stably expressing the control vector, SIRT5 WT, and SIRT5 H158Y. SIRT5 WT overexpression significantly accelerated CRC tumorigenesis (revised Supplementary Figure 6a–c). Moreover, in the SIRT5 WT group, TKT activity in tumors showed a significant increase, while it remained low in the SIRT5 H158Y and control vector groups (revised Supplementary Figure 6d). Targeted metabolomics analysis of tumor lysates revealed that SIRT5 overexpression resulted in a significant increase in the levels of R5P, ATP, CDP, dCTP, and dGTP (revised Supplementary Figure 6e). Thus, these results supported that SIRT5 contributes to the malignant phenotype of CRC by activating TKT.

- Both cell lines the authors use in this study have oncogenic KRas mutations; is w metabolic effects the authors see in the study, or is this regulation of the PPP/TKT in cells intrinsic to SIRT5 regardless of the presence of additional driver mutations? Inclusion of experiments throughout the study in a CRC cell line with WT Kras (HT29,

SW48) or another cell line would shed light on the relevance/contribution of oncogenic driver mutations to the observed SIRT5-associated phenotypes. This is especially relevant to consider since studies in SIRT5 KO mice in the absence of additional challenge/mutations have not revealed any strong phenotypes.

Response: To address your feedback, we detected γ H2AX expression level in HT29 cells, a CRC cell line with WT KRAS, treated with SIRT5 siRNAs for 24, 36, 48, and 72 h. In line with the results for HCT116 cells, we found that SIRT5 silencing upregulated γ H2AX expression levels (revised Supplementary Figure 1a, left). An increase in the number of γ H2AX nuclear foci in HT29 cells after SIRT5 knockdown is shown in revised Supplementary Figure 1b, c. Besides, we performed EdU assay involving immunofluorescence and flow cytometry with HT29 cells after SIRT5 silencing and found that SIRT5 knockdown decreased the uptake rate of EdU and mean fluorescence intensity of EdU⁺ cells (revised Supplementary Figure 3a–d). Moreover, the inhibition of DNA synthesis induced by SIRT5 silencing was reversed in HT29 cells upon exogenous nucleoside supplementation (revised Supplementary Figure 3a–d). We then observed the effects of suppressing SIRT5 on cell cycle and apoptosis and found that exogenous supplementation with the four nucleosides alleviated SIRT5 silencing-induced cell cycle arrest (revised Supplementary Figure 3i, j) and apoptosis (revised Supplementary Figure 3k, l). Consistently, Western blotting revealed that the expression levels of cyclin D1 and cyclin D3 were downregulated, while those of cyclin E1 and cyclin A2 were upregulated in HT29 cells transfected with SIRT5 siRNAs, but these effects were restored on nucleoside supplementation (revised Supplementary Figure 3m). In addition, the levels of apoptosis indicators, including cleaved caspase 3, caspase 8, caspase 9, and PARP were downregulated in HT29 cells with SIRT5 silencing by exogenous nucleoside supplementation (revised Supplementary Figure 3m). These data supported that SIRT5 silencing-induced phenotypes was independent of the KRAS mutation in CRC cells.

- To more strongly establish the SIRT5-TKT axis as an attractive therapeutic target in CRC, the study would benefit from in vivo studies in a more cancer-relevant mouse

model than the xenograft studies shown in Figure 7. Either a GEMM with oncogenic driver mutations present (Apc, Kras) where SIRT5 could be knocked out or orthotopic injection of CRC tumor cells, would be a better model for the *in vivo* experiments.

Response: Thank you for these useful suggestions. To explore the role of the SIRT5–TKT axis on tumor behavior *in vivo*, we generated a surgical orthotopic mouse model by injecting luciferase-transfected HCT116 cells stably expressing NTC shRNA or SIRT5 shRNAs into the cecum of nude mice. A bioluminescence imaging system (see revised Figure 7a, b) was used to monitor tumor growth on a weekly basis, and the animals were sacrificed 4 weeks after tumor implantation. In comparison with the control group, both tumor volume and weight in the SIRT5-silencing group were significantly restrained (revised Figure 7c–e). Furthermore, we analyzed TKT activity in tumor tissues and found that TKT activity in SIRT5-knockdown tumors was decreased by 38% as compared with that in the control group (revised Figure 7f). Targeted metabolomics analysis of tumor lysates revealed that SIRT5 silencing resulted in a significant downregulation in the levels of R5P, dAMP, dGMP, AMP, UMP, CMP, and GMP (revised Figure 7g). In addition, Western blotting confirmed the stable knockdown of SIRT5 in the lysates of orthotopic tumors, and a significant change in TKT protein levels were not observed (revised Figure 7h). As expected, SIRT5 silencing considerably increased γ H2AX levels in orthotopic tumors (revised Figure 7h). Collectively, these results supported our *in vitro* data, validating that SIRT5 plays a key role in CRC tumorigenesis by activating TKT, and sufficient nucleotide levels are consequently maintained for DNA synthesis.

- The conclusions of the study regarding the relevance of SIRT5-mediated regulation of TKT activity in CRC would be strengthened by analyzing TKT activity from either GEMM tumor samples where SIRT5 was knocked out, or CRC patient tumor samples.

Response: According to your suggestion, we performed further analyses. As shown in revised Figure 7f, relative to the control group, TKT activity in SIRT5-knockdown tumors was decreased by 38% compared with that in the control tumors in an orthotopic mouse model of CRC. Western blotting confirmed the stable knockdown of SIRT5

in tumor lysates, and we did not observe a significant change in TKT protein levels (revised Figure 7h).

REFERENCES

6. Wang YQ, Wang HL, Xu J, Tan J, Fu LN, Wang JL, *et al.* Sirtuin5 contributes to colorectal carcinogenesis by enhancing glutaminolysis in a deglutarylation-dependent manner. *Nature communications* 2018, **9**(1): 545.
7. Nakagawa T, Lomb DJ, Haigis MC, Guarente L. SIRT5 Deacetylates carbamoyl phosphate synthetase 1 and regulates the urea cycle. *Cell* 2009, **137**(3): 560-570.
8. Park J, Chen Y, Tishkoff DX, Peng C, Tan M, Dai L, *et al.* SIRT5-mediated lysine desuccinylation impacts diverse metabolic pathways. *Molecular cell* 2013, **50**(6): 919-930.
9. Boren J, Ramos-Montoya A, Bosch KS, Vreeling H, Jonker A, Centelles JJ, *et al.* In situ localization of transketolase activity in epithelial cells of different rat tissues and subcellularly in liver parenchymal cells. *The journal of histochemistry and cytochemistry : official journal of the Histochemistry Society* 2006, **54**(2): 191-199.
10. Qin Z, Xiang C, Zhong F, Liu Y, Dong Q, Li K, *et al.* Transketolase (TKT) activity and nuclear localization promote hepatocellular carcinoma in a metabolic and a non-metabolic manner. *Journal of Experimental & Clinical Cancer Research* 2019, **38**(1).

Reviewer #3 (Remarks to the Author):

In the present manuscript, Wang and colleagues studied the role of sirtuin 5 (SIRT5) in CRC metabolism and proliferation. SIRT5 is a mitochondrial enzyme that catalyzes the removal of PTMs including acetylation, succinylation and malonylation in a NAD⁺-dependent manner. Here, different CRC models, including two CRC cell lines, a CRC xenograft mouse model as well as human colorectal tumor specimens were used to detail the role of SIRT5. Using siRNA-mediated knockdown of SIRT5, the authors showed reduced levels of ribose-5-phosphate (R5P) and, consequently, lower levels of dNTPs. This resulted in the formation of DNA damage and cell death induction in CRC cell lines, although the exact mechanism remains to be shown. Interestingly, the authors revealed a link between transketolase (TKT), an enzyme required in the non-oxidative pentose phosphate pathway, and SIRT5. They provided mechanistic evidence that SIRT5-mediated demalonylation of TKT promotes TKT activity and thus R5P production, which is necessary for the proliferation of CRC cells. These findings were translated into an in vivo CRC xenograft model, corroborating the in vitro results.

In general, this is an interesting study, which provides novel insights into the metabolic deregulation of CRC cells as potential target for CRC therapy. Nevertheless, I see some **major issues**, which are particularly important for the translational aspects and the scope of this study.

Specific comments:

1-Fig. 1: It is important to include p53-mutated CRC cells and normal, non-malignant colonocytes (see above).

Response: Thank you for such constructive feedback. As suggested, we detected γ H2AX levels after SIRT5 knockdown in the p53 mut CRC cell line HT29 and in the human normal colon epithelial cell line NCM460. Similar results were observed in HT29 cells. However, SIRT5 seemed to have a negligible effect on γ H2AX levels in NCM460 cells (revised Supplementary Figure 1a-c). Moreover, SIRT5 knockdown decreased

DNA synthesis in HT29 cells, which could be reversed by supplementation with the four nucleosides (revised Supplementary Figure 3a–d). Next, we investigated the effects of exogenous nucleoside supplementation on cell cycle and apoptosis of HT29 cells after SIRT5 silencing. We found that exogenous nucleoside supplementation alleviated SIRT5 silencing-induced cell cycle arrest (revised Supplementary Figure 3i, j) and apoptosis (revised Supplementary Figure 3k, l). Consistently, Western blotting revealed that the levels of apoptosis indicators, including cleaved caspase 3, caspase 8, caspase 9, and PARP, and the DNA damage marker γ H2AX were downregulated in HT29 cells with SIRT5 silencing by exogenous nucleoside supplementation (revised Supplementary Figure 3m). In addition, the expression levels of cyclin D1 and cyclin D3 were downregulated, while those of cyclin E1 and cyclin A2 were upregulated in HT29 cells transfected with SIRT5 siRNAs, but these effects were restored by nucleoside supplementation (revised Supplementary Figure 3m). Altogether, it appears that SIRT5 silencing-induced DNA damage results due an insufficient pool of nucleotides, subsequently affecting cell cycle arrest and apoptosis in CRC cells instead of non-malignant colonocytes.

2-Fig. 1B: please provide images with higher magnification to assess nuclear γ H2AX staining. Please note that both in the right panel (HCT116) as well as in the left panel (LoVo) apoptotic cells are clearly visible (condensed nuclei in DAPI channel). It is known that γ H2AX could also arise due to apoptosis. Therefore, it is recommended to check the contribution of apoptosis-mediated γ H2AX formation to the overall γ H2AX levels by using pan-caspase inhibitors.

Response: To address your feedback, we repeated the pertinent experiment. Immunofluorescence staining of γ H2AX (Ser139) was performed. HCT116 (left) and LoVo (right) cells were pretreated with the pan-caspase inhibitor z-VAD (50 μ M) for 24 h. As shown in revised Figure 1b-c, after pretreatment with capsase inhibitors z-VAD-FMK (z-VAD), γ H2AX nuclear foci was increased in SIRT5-knockdown cells, indicating that SIRT5 knockdown itself, but not through apoptosis, could also cause significant DNA damage.

3-Fig. 1D and E: The authors showed the formation of DNA strand breaks using the alkaline Comet assay. However, what is the underlying mechanism? It is strongly recommended to analyze further markers of replication stress (e.g. RPA foci or DNA fiber assay). Including HU as positive control would also make sense (see comment above).

Response: Thank you for this excellent suggestion. The alkaline comet assay is used to detect DNA strand breaks. The relaxed loops of damaged DNA extend to the anode under the action of an electric field to form a comet-shaped structure (i.e., a tail), which can be seen by fluorescence staining. Herein our data suggested that SIRT5 knockdown caused DNA strand breaks. Persistent replication stress, including DNA lesions and nucleotide insufficiency, can stall replication forks and/or replication-fork collapse, resulting in the accumulation of single-stranded DNA gaps or DNA double-strand breaks^{11, 12}. To investigate if the mechanism underlying DNA strand breaks is related to replication stress, we measured the protein levels of pRPA, which is a marker of replication stress and is accumulated in nuclear speckles at sites of stalled replication¹³, using immunofluorescence (revised Figure 1f, g) and Western blotting (revised Figure 1h), with HU as a positive control. We found that pRPA levels were increased in nuclear speckles after SIRT5 knockdown. However, nucleoside supplementation decreased pRPA expression levels (revised Figure 2h). These data supported that replication stress might be responsible for SIRT5 silencing-induced DNA strand breaks. We have accordingly modified pertinent sections in the revised manuscript.

4-Fig. 1F: It is not mentioned after which time point the DDR markers were analyzed. Furthermore, a time course experiments similar to fig. 1a would help to elucidate the time kinetics (i.e. transient or persistent DDR activation) and provide some clues on the mechanism.

Response: Thanks for this suggestion. The DDR markers were analyzed in SIRT5 siRNAs-treated HCT116 and LoVo cells for 48 h (revised Figure 1i). We performed a time-course experiment and found that pATM, pATR, pCHK1, and pCHK2 expression levels were increased after SIRT5 silencing for 24, 36, 48, and 72 h (Supplementary

Figure 5 for reviewers, see below), implying that DDR activation is associated with response to DNA damage.

Supplementary Figure 5 for reviewers

Supplementary Figure 5 for reviewers. Representative Western blots showing DDR pathway upregulation at different timepoints in SIRT5-depleted HCT116 cells compared with control cells. β -actin was used as a loading control.

5-Fig. 2C and D: an alkaline Comet assay would strengthen the notion that supplementation with nucleosides can reverse DNA damage.

Response: To address this comment, we performed the alkaline comet assay to observe whether nucleoside supplementation affected SIRT5 silencing-induced DNA damage. As shown in revised Figure 2f, g, nucleoside supplementation reversed DNA damage observed in response to SIRT5 silencing.

6-Fig.3: please include normal colonocytes and p53-mutated CRC cell lines to allow for a more general conclusion and to address the point of tumor selective cell death. DNA damage typically engages p53 as a major trigger of apoptotic cell death, thus p53-mutated cells have to be considered.

Response: Thank you for these helpful comments. p53-mutated HT29 cells showed significant cell apoptosis after SIRT5 knockdown, indicating that SIRT5 silencing-induced apoptotic cell death was independent of p53 (revised Supplementary Figure 3k, l). In addition, we identified that SIRT5 had a negligible little effect on DNA damage in the human normal colon epithelial cell line NCM460 (revised Supplementary Figure 1a).

7-Fig. 1-3: A pharmacological approach using **SIRT5 inhibitor 1** could nicely cross-validate key findings obtained by SIRT5 knockdown.

Response: In response to this suggestion, we determined the effects of SIRT5 inhibitor 1, a newly synthesized specific human SIRT5 deacylase inhibitor, on DNA synthesis, cell cycle arrest and apoptosis, TKT activity. HCT116 cells were treated with different concentrations of SIRT5 inhibitor 1 to examine its effect on TKT activity. We found that SIRT5 inhibitor 1 inhibited TKT activity in a concentration-dependent manner (revised Supplementary Figure 5i). The optimal concentration of SIRT5 inhibitor 1 to inhibit TKT activity was 50 μ M. Impaired DNA synthesis, cell cycle arrest, and increased DNA damage and apoptosis induced by SIRT5 inhibitor 1 could be reversed by supplementation with the four nucleosides (revised Supplementary Figure 4a–g).

8-Fig. 4F: Based on the microscopy images, both TKT and SIRT5 show cellular pan-staining and no organelle-specific staining (i.e. SIRT5 in mitochondria). Could the authors please comment on that? Furthermore, a technical question came up: are these thin optical sections or z-stacks with maximum intensity projection shown here? This might critically impact the conclusions.

Response: We would like to thank you for these insightful comments, which greatly improve our manuscript. We optimized dilutions of the two antibodies and re-detected SIRT5 and TKT localization in CRC cells by immunofluorescence (revised Figure 4f, g). The obtained results confirmed that SIRT5 and TKT have strong co-localization. Thin optical sections with optimal intensity were used in this study.

9-Fig. 4m: the authors mentioned that the observed lower γ H2AX levels in HCT116 cells with WT SIRT5 overexpression after CPT treatment could be a result of increased DSB repair due to higher dNTP pools. This is very speculative and should be removed (or addressed experimentally). The expectation would rather be the opposite: given that SIRT5 overexpression causes higher dNTP levels and thus promotes DNA replication, the effects of CPT should be stronger. The TOPO I poison CPT is active in replicating cells, causing a collision of the replication fork with the cleavage complex formed by DNA, TOPO I and CPT.

Response: We agree that SIRT5 WT overexpression could trigger replication stress after CPT treatment. Therefore, Figure 4m has now been omitted.

10-Fig. 7e: Another genotoxicity marker, such as p53, should be detected to corroborate the notion that SIRT5 k.o. causes increased levels of DNA damage in tumor tissue.

Response: We assessed the protein level of p53 after SIRT5 silencing in tumor lysates derived from xenografts. Western blotting showed that SIRT5 knockdown significantly increased the expression levels of p53, which was reversed by TKT overexpression (revised Figure 7n).

11-Fig. 7f: please perform quantitative evaluation of γ H2AX levels either by WB or IHC

Response: Thank you for this helpful suggestion. First, the grayscale intensities of protein bands were quantified using ImageJ (National Institute of Health). We found that γ H2AX levels in SIRT5-knockdown tumors were about 8 times higher than those in control tumors. On assessing tumor lysates derived from xenografts, we observed that this effect was reversed following TKT overexpression (Supplementary Figure 6a for reviewers, see below). Second, staining intensity score was evaluated on a scale of 0–3: 0 = no staining, 1 = weak staining, 2 = moderate staining, and 3 = strong staining. Further, staining extent score indicated the percentage of positively stained cells (0 = 0%–5%, 1 = 6%–25%, 2 = 26%–50%, 3 = 51%–75%, and 4 = 76%–100%). To derive the final score (γ H2AX expression), staining intensity and extent scores were multiplied. As expected, samples with SIRT5 knockdown displayed strong staining of

γ H2AX (Supplementary Figure 6b for reviewers, see below).

Supplementary Figure 6 for reviewers

Supplementary Figure 6 for reviewers. a, b γ H2AX protein was extracted from tumor lysates derived from xenografts, and after different treatments, γ H2AX levels were determined by Western blotting and immunohistochemistry. Band intensity of γ H2AX was calculated using ImageJ (a). Semiquantitative assessment of immunohistochemical staining of γ H2AX (b).

12-Fig. 7h: Can the authors please detail on the antibody validation and the evaluation of the stainings? Further, how was distinguished between staining of tumor cells and tumor stroma?

Response: Thank you for these queries. An antibody was considered to be validated if it produced a band (or bands) of the expected molecular weight(s) for the target protein¹⁴. For SIRT5 and γ H2AX antibody validation (catalog #9718), SIRT5 and γ H2AX expression levels were detected after treating HCT116 cells with SIRT5 siRNAs and HU (positive control), respectively. Western blotting indicated that a band corresponding to the predicted size of SIRT5 (approximately 33 kDa) and γ H2AX (approximately 15 kDa) was amplified (Supplementary Figure 7a, b for reviewers). SIRT5 immunohistochemistry was also used to validate antibody reliability, which can be available on the Human Protein Atlas (<http://www.proteinatlas.org>). Moreover, immunohistochemical analysis for γ H2AX of paraffin-embedded human breast carcinoma can

be seen on the website of Cell Signaling Technology. Experimental conditions for immunohistochemistry were optimized, and experiments were re-performed (revised Figure 8a).

Protein expression levels were quantified using a visual grading system based on the extent of staining (percentage of positive tumor cells) and intensity of staining. The expression levels of SIRT5 and TKT were assessed according staining intensity and extent. Staining intensity score was evaluated on a scale of 0–3: 0 = no staining, 1 = weak staining, 2 = moderate staining, and 3 = strong staining. Further, staining extent score indicated the percentage of positively stained cells (0 = 0%–5%, 1 = 6%–25%, 2 = 26%–50%, 3 = 51%–75%, and 4 = 76%–100%). To derive the final score (protein expression), staining intensity and extent scores were multiplied. For further analysis, the multiplication product (the corresponding score) was used to define the cutoff value for different protein expression levels. In the correlative study, the expression levels of relevant proteins were classified into two categories.

To distinguish staining of tumor cells and tumor stroma, sections of cancerous tissues were exposed to hematoxylin and eosin (H&E) staining. Differentiation was based on pathologic criteria. Tumor cells are characterized by pleomorphism, hyperchromatic nuclei, high nuclear-to-cytoplasmic ratio, and presence of giant cells (Supplementary Figure 7c for reviewers, see below). These tissues were assessed by pathologists at Renji Hospital, Shanghai Jiao Tong University, School of Medicine.

Supplementary Figure 7 for reviewers

Supplementary Figure 7 for reviewers. a, b Cell lysates obtained from HCT116 cells exposed to different treatments were denatured and separated by 10% SDS-PAGE, transferred to a nitrocellulose membrane, and blotted with rabbit antibodies against SIRT5 (a) and γ H2AX (b), which led to the generation of a specific band of the expected molecular weight. No other bands of unexpected molecular weights were seen. c Representative images of H&E staining of tumor tissues derived from patients with CRC.

M&M part:

- Cell culture: how was mycoplasma contamination tested?

Response: MycoBlue Mycoplasma Detector (Vazyme Biotech, Nanjing, China) was used for this purpose.

- Western blots: how was chemoluminescence measured (imager?) and was densitometric evaluation of blots performed?

Response: We used the ChemiDoc™ MP Imaging System for imaging gels and blots. Immunoblots were quantified by Image J software (Image J, NIH). Such information has now been included in the revised manuscript.

- Confocal microscopy: z-stacks or single optical sections? Which cLSM was used?

Response: We used the Zeiss LSM 710 confocal microscope (Carl Zeiss) fitted with a 63× oil immersion objective. Thin optical sections with optimal intensity were assessed.

- Comet-Assay: DNA staining used? How many comets were assessed per slide/sample?

Response: We used the Vista Green DNA Dye. Tail moments of at least 100 cells per slide per sample were analyzed using CometScore.

- IHC: how were the tissues collected and fixed? FFPE tissue and microtome sectioning or snap-frozen tissue and cryostat sectioning?

Response: Tissues were obtained from patients with CRC who underwent surgery and colonoscopy at Renji Hospital, Shanghai Jiao Tong University, School of Medicine. For immunohistochemistry, 6-µm formalin-fixed paraffin-embedded sections were used.

- In vivo models: source and nature of adenoviruses targeting SIRT5 and TKT?

Response: The adenovirus NTC shRNA, adenovirus SIRT5 shRNAs, as well as TKT overexpression adenovirus were constructed by Obio Technology Company (Shanghai, China). We used a replication-defective human adenovirus type 5 (AdV5) vector to deliver shRNAs targeting SIRT5 into cells. We have added this information to the revised manuscript and also included relevant data.

- Source of human normal and tumor tissue? Ethics statement? Animal experiments approval number?

Response: Tissues were obtained from patients with CRC who underwent surgery and colonoscopy at Renji Hospital, Shanghai Jiao Tong University, School of Medicine. All animal studies were conducted according to the guidelines published by the Animal Ethics Committee of Renji Hospital, Shanghai Jiao Tong University, School of Medicine. Written informed consents were obtained from all participants. All study protocols were performed in accordance with the 1975 version of the Declaration of Helsinki. The approval number for animal experiments is 2019-0024.

- Plasmids and constructs used?

Response: All plasmids, including TKT WT/K281R/K282R/K283R plasmids, were constructed by Generay Technologies (Shanghai, China). We have added this information in the revised manuscript.

- ROS assay and 8-OH-dG detection performed in Fig. S1: no information provided!

Response: We have addressed this issue in revised Supplementary Information. Intracellular reactive oxygen species (ROS) levels were detected in control and SIRT5 siRNA-transfected cells at 48 h, followed by treatment with 10 μ M DCFDA probe (Abcam) for 30 min. Cells treated with 10 μ M hydrogen peroxide (H_2O_2) for 30 min were used as a positive control. ROS production was measured by flow cytometry. SIRT5 silencing-induced ROS levels were measured by monitoring the increase in fluorescence. Values represent mean \pm SD of three independent experiments.

Levels of 8-hydroxy-2'-deoxyguanosine (8-OH-dG, a marker of DNA/RNA oxidative

damage) were measured in HCT116 and LoVo cells at 48 h post-transfection with control or SIRT5 siRNAs using an ELISA kit (Abcam, ab201734, USA). Cells treated with 10 μ M H₂O₂ for 30 min were used as a positive control. Briefly, DNA was purified from cell lysates using a commercial extraction kit. The sample was then digested with nuclease P1 and incubated with alkaline phosphatase for 30 min at 37°C. The prepared 8-OH-dG standard, zero standard, and digested DNA samples were added to a plate with diluted 8-OH-dG antibody, followed by incubation at room temperature for 1 h. The TMB substrate reaction was allowed to proceed in the dark at room temperature for 30 min, and absorbance at 450 nm was then measured using an ELISA plate reader. Relevant data have been included in Supplementary Information.

REFERENCES

11. Alexander JL, Barrasa MI, Orr-Weaver TL. Replication fork progression during re-replication requires the DNA damage checkpoint and double-strand break repair. *Current biology : CB* 2015, **25**(12): 1654-1660.
12. Cortez D. Preventing replication fork collapse to maintain genome integrity. *DNA repair* 2015, **32**: 149-157.
13. Robison JG, Elliott J, Dixon K, Oakley GG. Replication protein A and the Mre11.Rad50.Nbs1 complex co-localize and interact at sites of stalled replication forks. *The Journal of biological chemistry* 2004, **279**(33): 34802-34810.
14. Major SM, Nishizuka S, Morita D, Rowland R, Sunshine M, Shankavaram U, *et al.* AbMiner: a bioinformatic resource on available monoclonal antibodies and corresponding gene identifiers for genomic, proteomic, and immunologic studies. *BMC bioinformatics* 2006, **7**: 192.

REVIEWER COMMENTS

Reviewer #1 (Remarks to the Author):

The authors have satisfactorily addressed all my concerns. They carried out several experiments to strengthen their scientific findings and corrected the remaining minor issues for the manuscript.

I have no further comments but to suggest formal approval of the paper.

Reviewer #3 (Remarks to the Author):

The authors made great efforts during the revision stage and carefully addressed all points of concern. The manuscript has significantly improved and is now acceptable for publication in Nature Communications.

Reviewer #4 (Remarks to the Author):

N-cadherin upregulation mediates adaptive radioresistance in glioblastoma

Summary - In this manuscript, Wang et al present a very significant amount of experimental tissue culture and xenograft in vivo data suggesting that SIRT5-TKT-axis, which targets PPP it least in some significant part, leads to changes in the cellular DNA damage response in CRC model systems and in human data sets. It is also proposed that TKT activity is a significant mechanism, as regulated by SIRT5, in this process. All of the experiments were done with SIRT5 knockdown shRNA to show that reduced levels of R5P and dNTPs leading to increased DNA damage and cell death. Overall, this is a novel and potentially important manuscript in a significant amount of new data and it would be well received in Sirtuin and DNA damage and repair communities. The authors have also provided very significant amounts of new data in the revised manuscript and this should be strongly considered in making a recommendation. However, the manuscript still has some weakness that lead to a decrease in scientific rigor.

(i) The study does still not determine establish how SIRT5, a mitochondrial enzyme in mammalian cells, regulates a cytosolic process including DNA repair. In this regard, I know of no publication showing retrograde protein transport out of the mitochondrial and some of the findings are in non-mitochondrial compartments of the cell. The authors do not demonstrate how signaling is going on between the mitochondrial and other parts of the cell and determine this will be difficult.

(ii) The authors rely upon γ H2X and comet assays to measure DNA damage and I think more rigorous methods should be employed to determine the specific DNA damage. These types of studies are quite indirect and more rigorous and specific experiments should be done than measure biomarkers for DNA damage, which is descriptive. Also, the data suggesting replicative stress is also descriptive.

(iii) The mechanism for the regulation of TKT is still not complete. Identifying the specific lysine is an important step but not experiments are presented to address a cause and effect between changes in TKT enzymatic activity and DNA repair. The transformation assays are intriguing but they still not address this important part of the manuscript.

(iv) The question of the different isoforms from the initial reviewers has not been addressed and showing that TKT is in the mitochondrial really does not address this. And as stated in (i) the signaling between the mitochondrial and the rest of the cell is still not really address in this revised version of the manuscript as is the question of the different isoforms.

(v) The co-localization experiments are well done but to rigorously show that TKT is in the mitochondrial gold labeled EM experiments need to be done.

(vi) As suggested by one of the reviewers a GEMM tumor samples, which I assume would be SIRT5 knockout, were not done and knockdown cell lines still do not address this very important issue from the reviewer. This experiments could be done by different means since all of the reagents are readily available to make a compound mouse that forms tumors in a loss of Sirt5 background.

(vii) While some of the data very strongly supports parts of the conclusions other parts of the manuscript still are not scientifically complete.

(viii) In conclusion, this manuscript presents important and new experimental data for the mitochondrial sirtuin, SRIT5 that will positively affect the field. However, the issue of rigor and

scientific depth are still a bit lacking. In addition, while some of the data is very strongly and clearly supports the conclusions, other parts of the manuscript are still not scientifically complete.

Point by point response to reviewers' comments

Reviewer #1 (Remarks to the Author):

The authors have satisfactorily addressed all my concerns. They carried out several experiments to strengthen their scientific findings and corrected the remaining minor issues for the manuscript.

I have no further comments but to suggest formal approval of the paper.

Response: We sincerely appreciate anonymous reviewers for their thoughtful comments and efforts that helped us to improve the quality of the manuscript.

Reviewer #3 (Remarks to the Author):

The authors made great efforts during the revision stage and carefully addressed all points of concern. The manuscript has significantly improved and is now acceptable for publication in Nature Communications.

Response: We really thank the reviewers for helping us improve our manuscript.

Reviewer #4 (Remarks to the Author):

N-cadherin upregulation mediates adaptive radioresistance in glioblastoma

Summary - In this manuscript, Wang et al present a very significant amount of experimental tissue culture and xenograft in vivo data suggesting that SIRT5-TKT-axis, which targets PPP it least in some significant part, leads to changes in the cellular DNA damage response in CRC model systems and in human data sets. It is also proposed that TKT activity is a significant mechanism, as regulated by SIRT5, in this process. All of the experiments were done with SIRT5 knockdown shRNA to show that reduced levels of R5P and dNTPs

leading to increased DNA damage and cell death. Overall, this is a novel and potentially important manuscript in a significant amount of new data and it would be well received in Sirtuin and DNA damage and repair communities. The authors have also provided very significant amounts of new data in the revised manuscript and this should be strongly considered in making a recommendation. However, the manuscript still has some weakness that lead to a decrease in scientific rigor.

(i) The study does still not determine establish how SIRT5, a mitochondrial enzyme in mammalian cells, regulates a cytosolic process including DNA repair. In this regard, I know of no publication showing retrograde protein transport out of the mitochondrial and some of the findings are in non-mitochondrial compartments of the cell. The authors do not demonstrate how signaling is going on between the mitochondrial and other parts of the cell and determine this will be difficult.

Response: Thank you for such constructive comments. It has been widely reported that SIRT5 is found in both mitochondria and the cytosol^{1,2}, but the subcellular location of TKT is unknown. We found that SIRT5 and TKT were in both mitochondrial and cytoplasmic fractions using subcellular fractionation (Supplementary Figure5j). Although TKT is mostly distributed in the cytoplasm, confocal assay (Supplementary Figure 1a, b for reviewers) and gold labeled EM experiments (Supplementary Figure5k) indicated that TKT was also present in mitochondria. As immunofluorescent staining (revised Fig. 4f, g) and co-immunoprecipitation assay (Fig. 6c, d) revealed a strong co-localization between SIRT5 and TKT in HCT116 and LoVo cells, we further found that there is an interaction between SIRT5 and TKT in both cytosolic and mitochondrial fractions of HCT116 cells (Supplementary Figure 5l). Previous studies have demonstrated that SIRT5 has sufficient deacetylase ability to regulate metabolic processes in both mitochondria and cytoplasm. Nakagawa *et al.* reported that SIRT5 deacetylates and desuccinylates **the mitochondrial enzyme, carbamoyl phosphate synthetase 1 (CPS1)**, and regulates the urea cycle^{3, 4}. Another study revealed that SIRT5 increased the noncanonical use of glutamine via activating Asp aminotransferase (GOT1)⁵, **the enzyme responsible for aspartate transamination in the cytosol**. In line with this, we found that lysine malonylation levels of TKT were elevated in both cytosolic and mitochondrial fractions of SIRT5-silenced HCT116

cells (Supplementary Figure 5I), showing that TKT may be demalonylated by SIRT5 in both mitochondrial and cytoplasmic fractions. In the revised manuscript, corrections are marked in red.

It is well documented that in eukaryotic cells a single protein can be located in more than one subcellular compartment, such as fumarase, which is present in the mitochondria, cytosol, and nucleus⁶. The pyruvate dehydrogenase complex (PDC) is a mitochondrial metabolic complex that regulates acetyl-CoA production. PDC has been shown to translocate from mitochondria to the nucleus by using a vesicular pathway analogous to mitochondrial-derived vesicles (MDVs)⁷. MDVs can segregate their cargo to peroxisomes or lysosomes and have been recently implicated in the delivery of mitochondrial antigens to MHC Class I⁸. We fully agree that elucidation of the mechanism for how TKT signaling is going on between the mitochondria and the cytoplasm will be interesting but challenging, which deserves huge work in the future.

Supplementary Figure 5j

Supplementary Figure 1a for reviewers

Supplementary Figure 1b for reviewers

Supplementary Figure 5k

Supplementary Figure 5l

Supplementary Figure 5j. Endogenous TKT and SIRT5 were detected in mitochondrial (Mito) and cytoplasmic (Cyto) fractions of HCT116 cells by immunoblot analysis.

Supplementary Figure 1a, b for reviewers. Endogenous TKT was partially localized to the mitochondria in CRC cells (a) and tissues (b). Immunofluorescence staining for TKT (red) and Mito-Track (green); yellow in the merged images indicates the co-localization between TKT and mitochondria. Scale bar in b = 5 μm and in d = 50 μm.

Supplementary Figure 5k. Images showing that the gold particle-labeled TKT was located in the mitochondria and cytoplasm of HCT116 cells using transmission electron

micrographs. White arrows indicate TKT protein. Scale bar, 200 nm.

Supplementary Figure 5I. Western blot analysis for lysine malonylation (MalK) levels of TKT in cytosolic or mitochondrial fractions derived from control and SIRT5-silenced HCT116 cells. The purity of subcellular fractionation was confirmed by western blot for the cytosolic protein β -actin and the mitochondrial protein cox IV.

(ii) The authors rely upon γ H2AX and comet assays to measure DNA damage and I think more rigorous methods should be employed to determine the specific DNA damage. These types of studies are quite indirect and more rigorous and specific experiments should be done than measure biomarkers for DNA damage, which is descriptive. Also, the data suggesting replicative stress is also descriptive.

Response: Thank you for your constructive comments. Our results showed that SIRT5 knockdown caused significant DNA damage using alkaline comet assay and γ H2AX. Further, we demonstrated that SIRT5 silencing significantly decreased nucleotide pool (Fig. 2a, b and Supplementary Fig. 2b), one important contributor of replication stress, which can induce DNA damage⁹, and exogenous supply of nucleosides abolished SIRT5 silencing-induced DNA damage (Fig. 2c–g). It's well reported that the ATR kinase, primarily responds to replication stress, where it is recruited during S phase through replication protein A (RPA) to stalled replication forks. Accordingly, we measured the protein levels of pRPA and pATR, and found that both of them were elevated after SIRT5 knockdown (Fig. 1f–h and Fig. 1i). To address your concern, we used DNA fiber assay to directly examine the impact of SIRT5 knockdown on DNA replication forks and found that SIRT5 deficiency slowed down the progression of replication forks (Supplementary Fig. 1d, e). Intriguingly, supplementation with nucleosides restored such changes (Supplementary Fig. 2f). Collectively, these results confirmed that replication stress induced by deficient nucleotides is accountable for SIRT5 knockdown-mediated DNA damage.

Supplementary Figure 1d,e. Control and SIRT5-depleted HCT116 cells were pulsed using labeling scheme. Representative fiber images for indicated samples were shown (d). CldU and IdU lengths were measured and converted to kilo bases ($1 \mu\text{m} = 2.59 \text{ Kb}$), and fork speed was plotted as histograms (e). At least 300 fibers were recorded from 3 experimental replicates. Statistical significance was calculated using two-tailed unpaired t-test. *** $p < 0.001$, **** $p < 0.0001$. **Supplementary Figure 2f.** Fork speed was performed in SIRT5-silenced HCT116 cells after supplementation with exogenous nucleosides.

(iii) The mechanism for the regulation of TKT is still not complete. Identifying the specific lysine is an important step but not experiments are presented to address a cause and effect between changes in TKT enzymatic activity and DNA repair. The transformation assays are intriguing but they still not address this important part of the manuscript.

Response: To address a cause and effect between changes in TKT enzymatic activity and DNA damage, vector control and SIRT5 WT transfected cells were transfected with control siRNA and TKT siRNAs, subsequently treated with $20\mu\text{M}$ 5-fluorouracil (5-FU), a DNA damaging agent, for 48h. We found that in comparison with the control vector, DNA damage in cells overexpressing SIRT5 was significantly decreased after 5-FU treatment, which was reversed by the knockdown of TKT (Figure 5d). Consistently, treatment with TKT inhibitor oxythiamine (OT) also blocked SIRT5-induced protection from DNA damage in CRC cells (Figure 5e). Collectively, these results suggested that SIRT5 could protect CRC cells from DNA Damage by increasing levels of nucleotide pool via activating TKT.

Figure 5d

Figure 5e

Figure 5d, e. Immunoblotting of γ -H2AX in HCT116 and LoVo cells stably expressing the control vector and SIRT5 WT, followed by treatment with TKT siRNAs (d) or OT (20 μ M, e). 20 μ M 5-FU was used as a DNA-damaging agent.

(iv) The question of the different isoforms from the initial reviewers has not been addressed and showing that TKT is in the mitochondrial really does not address this. And as stated in (i) the signaling between the mitochondrial and the rest of the cell is still not really address in this revised version of the manuscript as is the question of the different isoforms.

Response: We apologize for making you confused. To the best of our knowledge, although cDNAs of all SIRT5 isoforms were readily detected in multiply tissues according to the EST database¹⁰, little research has focused on the functions of specific SIRT5 subtypes. Matsushita et al reported that SIRT5iso1 is localized in both mitochondria and cytoplasm, whereas the SIRT5iso2 is mainly localized in mitochondria¹. However, other researchers came to a different conclusion that SIRT5iso1-3 were mitochondria-localized, while SIRT5iso4 is mainly localized in cytoplasm¹⁰. Regarding the activity of different SIRT5 isoforms, SIRT5iso2-4 have shown much lower deacylase activity compared with SIRT5iso1¹⁰. The subcellular localization and functions of other SIRT5 isoforms have not been reported. Almost studies about metabolic reprogramming of SIRT5, containing glycolysis^{11, 12}, restricting TCA cycling², and maintaining electron transport^{2, 13}, haven't focused on the specific isoform of SIRT5. In this study, we did not focus on a specific isoform of SIRT5. SIRT5 siRNAs used herein were targeted at the conserved sequences of SIRT5 iso1-7.

(v) The co-localization experiments are well done but to rigorously show that TKT is in the mitochondrial gold labeled EM experiments need to be done.

Response: Thank you for the valuable suggestion. Subcellular fractionation and confocal immunofluorescent localization experiments indicated that TKT was in both mitochondrial and cytoplasmic fractions. To make the conclusion more rigorous, we further used immunogold labeling and transmission electron microscopy techniques to immunolocalize endogenous TKT at the subcellular level. We observed that TKT was localized in both mitochondria and cytoplasm of HCT116 and LoVo cells (Supplementary Figure 5k), consistent with subcellular fractionation findings.

Supplementary Figure 5k

Supplementary Figure 5k. Images showing that the gold particle-labeled TKT was located in the mitochondria and cytoplasm of HCT116 cells using transmission electron micrographs. White arrows indicate TKT protein. Scale bar, 200 nm.

(vi) As suggested by one of the reviewers a GEMM tumor samples, which I assume would

be SIRT5 knockout, were not done and knockdown cell lines still do not address this very important issue from the reviewer. These experiments could be done by different means since all of the reagents are readily available to make a compound mouse that forms tumors in a loss of Sirt5 background.

Response: Thanks for your suggestions. Reviewer #2 suggested that either a GEMM with oncogenic driver mutations present (Apc, Kras) where SIRT5 could be knocked out or orthotopic injection of CRC tumor cells, would be a better mouse model. We fully agree that both the CRC orthotopic mouse model and the GEMM models are useful for enhancing our understanding of cancer development. Many studies using an orthotopic mouse model to evaluate the role of one specific gene during tumor progression^{14, 15}. Given that the orthotopic model of CRC in mice is very similar to what happens on human tissue, replicating human cancer with high reliability^{16, 17}, we generated a surgical orthotopic mouse model to show changes in nucleotide metabolism of human CRC tumor cells with stable knockdown of SIRT5. Moreover, the orthotopic CRC model is more convenient for tumor monitoring and real-time visualization. The results indicated that SIRT5 played an important role in CRC tumorigenesis by activating TKT, which maintains sufficient nucleotides required for DNA synthesis.

(vii) While some of the data very strongly supports parts of the conclusions other parts of the manuscript still are not scientifically complete.

Response: Thanks for your help with improving our manuscript. Our in vitro and in vivo studies revealed SIRT5 silencing induced DNA damage by analyzing the γ H2AX, comet assay and TUNNEL assay. To figure out the underlying mechanism, we performed mass spectrometry (MS) and ¹³C-based metabolic flux analyses. We found that SIRT5 knockdown reduced R5P production by inhibiting the non-oxidative PPP. Supplementation with nucleosides restored replication stress (pRPA foci and DNA fiber assay), DNA damage, cell cycle arrest, and cell apoptosis induced by SIRT5 silencing. Specifically, SIRT5 activated TKT through demalonylation at lysine 281, thereby sustaining the nucleotide pools through the non-oxidative PPP. This interaction between SIRT5 and TKT were supported by co-immunoprecipitation, confocal immunofluorescent, and point-mutation

experiments. Besides, we observed that TKT was localized in both mitochondria and cytoplasm of HCT116 cells using subcellular fractionation and gold-labeled EM experiments. We further found that TKT could be demalonylated by SIRT5 in both mitochondrial and cytoplasmic fractions. To explore the role of the SIRT5-TKT axis on human tumor behavior and the related metabolic changes in vivo, we generated a surgical orthotopic mouse model with SIRT5 shRNA, reaching similar conclusions that SIRT5 could increase R5P generation and support nucleotide synthesis by activating TKT, thereby protecting DNA damage in CRC, which is also confirmed by subcutaneous xenograft tumor model.

(viii) In conclusion, this manuscript presents important and new experimental data for the mitochondrial sirtuin, SIRT5 that will positively affect the field. However, the issue of rigor and scientific depth are still a bit lacking. In addition, while some of the data is very strongly and clearly supports the conclusions, other parts of the manuscript are still not scientifically complete.

Response: We really appreciate the reviewer's advice for helping us improve our manuscript. According to your valuable suggestion, we have performed related experiments. This study revealed a new mechanism of how SIRT5, an important enzyme of metabolic reprogramming, protects CRC cells from DNA damage to promote CRC tumorigenesis via non-canonical post-translational modifications (PTMs). Although the results of our study are quite encouraging, there still has some problems that need to be studied in depth. Collectively, our data provide a better understanding of the close interaction among SIRT5, cell metabolism, and DNA damage, and also suggest that SIRT5 can serve as a promising target for CRC treatment.

REFERENCES

1. Matsushita N, *et al.* Distinct regulation of mitochondrial localization and stability of two human Sirt5 isoforms. *Genes Cells* **16**, 190-202 (2011).
2. Park J, *et al.* SIRT5-mediated lysine desuccinylation impacts diverse metabolic pathways. *Mol Cell* **50**, 919-930 (2013).

3. Nakagawa T, Lomb DJ, Haigis MC, Guarente L. SIRT5 Deacetylates carbamoyl phosphate synthetase 1 and regulates the urea cycle. *Cell* **137**, 560-570 (2009).
4. Du J, *et al.* Sirt5 is a NAD-dependent protein lysine demalonylase and desuccinylase. *Science* **334**, 806-809 (2011).
5. Hu T, *et al.* Metabolic Rewiring by Loss of Sirt5 Promotes Kras-Induced Pancreatic Cancer Progression. *Gastroenterology* **161**, 1584-1600 (2021).
6. Yogev O, Naamati A, Pines O. Fumarase: a paradigm of dual targeting and dual localized functions. *FEBS J* **278**, 4230-4242 (2011).
7. Sugiura A, McLelland GL, Fon EA, McBride HM. A new pathway for mitochondrial quality control: mitochondrial-derived vesicles. *EMBO J* **33**, 2142-2156 (2014).
8. Matheoud D, *et al.* Parkinson's Disease-Related Proteins PINK1 and Parkin Repress Mitochondrial Antigen Presentation. *Cell* **166**, 314-327 (2016).
9. Ragu S, Matos-Rodrigues G, Lopez BS. Replication Stress, DNA Damage, Inflammatory Cytokines and Innate Immune Response. *Genes (Basel)* **11**, (2020).
10. Du Y, Hu H, Hua C, Du K, Wei T. Tissue distribution, subcellular localization, and enzymatic activity analysis of human SIRT5 isoforms. *Biochem Biophys Res Commun* **503**, 763-769 (2018).
11. Lv XB, *et al.* SUN2 exerts tumor suppressor functions by suppressing the Warburg effect in lung cancer. *Scientific reports* **5**, 17940 (2015).
12. Wang F, *et al.* SIRT5 Desuccinylates and Activates Pyruvate Kinase M2 to Block Macrophage IL-1beta Production and to Prevent DSS-Induced Colitis in Mice. *Cell reports* **19**, 2331-2344 (2017).
13. Zhou L, *et al.* SIRT5 promotes IDH2 desuccinylation and G6PD deglutarylation to enhance cellular antioxidant defense. *EMBO reports* **17**, 811-822 (2016).
14. Wang Y, *et al.* LncRNA LINRIS stabilizes IGF2BP2 and promotes the aerobic glycolysis in colorectal cancer. *Mol Cancer* **18**, 174 (2019).
15. Pothuraju R, *et al.* Molecular implications of MUC5AC-CD44 axis in colorectal cancer progression and chemoresistance. *Mol Cancer* **19**, 37 (2020).
16. Hite N, *et al.* An Optimal Orthotopic Mouse Model for Human Colorectal Cancer Primary Tumor Growth and Spontaneous Metastasis. *Dis Colon Rectum* **61**, 698-705 (2018).
17. Liao HW, Hung MC. Intracaecal Orthotopic Colorectal Cancer Xenograft Mouse Model. *Bio Protoc* **7**, (2017).

REVIEWERS' COMMENTS

Reviewer #4 (Remarks to the Author):

The reviewers have done a excellent job of addressing my primary concern about how SIRT5 and TKT thought several different experiments that importantly include EM experiments showing TKT in the mitochondrial. When the EM data is added to the other methods to show this interaction it greatly improves the overall rigor of the manuscript and the authors have now addressed my concerns.

Point by point response to reviewers' comments

Reviewer #4 (Remarks to the Author):

The reviewers have done a excellent job of addressing my primary concern about how SIRT5 and TKT thought several different experiments that importantly include EM experiments showing TKT in the mitochondrial. When the EM data is added to the other methods to show this interaction it greatly improves the overall rigor of the manuscript and the authors have now addressed my concerns.

Response: Thanks for your positive and constructive comments. We appreciate the time and effort you have dedicated to improving our paper.